# Host dispersal relaxes selective pressures in rafting microbiomes and triggers successional changes

William S. Pearman [1,2,3] ✉, Grant A. Duffy[1], Robert O. Smith [1], Kim I. Currie[4], Neil J. Gemmell [2], Sergio E. Morales [3] & Ceridwen I. Fraser [1]

What little we know about how microbiomes change over the course of host dispersal has been gleaned from simulations or snapshot sampling of microbiomes of hosts undertaking regular, cyclical migrations. These studies suggest that major changes in both microbiome richness and turnover occur in response to long-distance movements, but we do not yet know how rare or sporadic dispersal events for non-migratory organisms might affect the microbiomes of their hosts. Here we directly examine the microbiomes of rafting seaweed, leveraging host genomic analyses, amplicon sequencing, and oceanographic modelling to study the impacts of ecological dispersal of hosts on their microbiomes. We find that once dislodged from coastal shores and adrift, kelp-associated microbial communities change profoundly—the core microbes found on attached kelp give way to a few abundant taxa and many rare taxa. Changes in microbial species richness and composition are strongly linked to variability in sea surface temperature rather than length of time spent rafting. These changes are associated with increased contributions of neutral processes shaping community assembly. These findings highlight the role of environmental predictability in triggering major community successional changes and challenge the importance of host selection in determining the microbiome.

Migratory and long-distance ecological movements can have profound impacts on host microbiomes, influencing community richness and diversity[1,2] of this ubiquitous and ecologically significant element of global biodiversity[3]. While we have gleaned insights into microbiome ecology from localized studies[1] and from host organisms whose movements are highly predictable (e.g., migrating whales[4]), direct observation and measurement of long-distance ecological movements is often challenging because dispersal events can be temporally and spatially disparate[5]. Longitudinal studies of microbiomes during dispersal are lacking, with most research examining discrete time points within the dispersal period[2,4,6] or conducting experiments removed

from the natural ecological context of the host[7,8]. The few studies attempting to grapple with these difficulties show that surface-associated microbiomes increase in richness in response to long-distance ecological movement, while internal communities tend to decline in richness[7]. With continuing environmental and climatic change necessitating species range shifts[9,10] and altering the timing and nature of long-distance ecological movements[11,12], understanding the effects of movement on microbiomes will be a critical component of predicting host response and survival in novel environments[13]. Furthermore, understanding how microbial communities shift in response to the dynamic environmental conditions experienced

[1]Department of Marine Science, University of Otago, Dunedin, New Zealand. [2]Department of Anatomy, School of Biomedical Sciences, University of Otago, Dunedin, New Zealand. [3]Department of Microbiology and Immunology, School of Biomedical Sciences, University of Otago, Dunedin, New Zealand. [4]National Institute of Water and Atmospheric Research, Dunedin, New Zealand. ✉e-mail: Wpearman1996@gmail.com

during movement could provide insight into microbiomes and host resilience against rapid environmental change more broadly. Indeed, environmental variability has been previously identified as a driver of community assembly in microbiomes, shaping the relative contribution of deterministic and stochastic processes[14].

A multi-tiered approach is needed to study the complexities of microbiomes as they change in space and time. For example, while 16S rRNA gene amplicon analyses can provide insight into microbial community assembly[15,16], it may miss the role of the host in facilitating microbial community change. Host biology and genetics have been repeatedly demonstrated to influence microbiome structure[17–19], including within the context of migratory birds[20]. Thus, study of the microbiome also requires understanding of, and expertise in, these host-centric factors. Furthermore, understanding host-microbiome dynamics in response to long-distance movement requires knowledge of source locations and the environments that hosts' microbiomes are exposed to while underway. Joint geographic and host-genomic data can act as proxies for the former[21,22], while modern-day remote sensing technologies and environmental modelling can be used to quantify the latter.

Buoyant macroalgae present an ideal system for addressing questions about host-movement-related changes to microbiomes, and they lend themselves to the multi-tiered approach required to disentangle the complexities of these questions. Macroalgae frequently achieve long-distance dispersal[21], their rafting routes can be inferred from oceanographic drift modelling[23], and surface oceanographic conditions can be measured using in situ and remote monitoring[24]. Furthermore, oceanic rafting can act as a vector for pathogenic microbes; rafts of the pelagic brown alga *Sargassum* have been implicated in the spread of pathogenic strains of *Vibrio* bacteria[6] and *Durvillaea antarctica* (southern bull kelp, rimurapa) has been identified as the likely transport vector of a microbial disease newly discovered in New Zealand[25] and other remote locations[26]. Indeed, at any one time it is estimated that there are up to 70 million rafts of *Durvillaea* floating in the Southern Ocean[27]; rafting events are frequent, can take place over tens of thousands of kilometres, and the source populations of rafting individuals can be reliably inferred using high-resolution genomic approaches[21,23].

*Durvillaea* is a genus of large, intertidal macroalgae, and individuals are occasionally detached from the coast by wave activity, sometimes then undergoing long-distance rafting[21,23,28–30]. The two buoyant species of *Durvillaea* which we focus on have been extensively studied[31–34], and a large amount of population genomic data exist for these taxa across their distributions[21,35–37]. As a result, beach-cast kelp and rafts can be confidently assigned to geographic sources[21,23]. Using such macroalgal rafts, we can construct a pseudo-time-series dataset by combining population genetic and oceanographic models to reconstruct a raft's likely journey. We can then compare the microbiomes of source populations to those of rafts to deconstruct the effects of host dispersal on microbiome structure.

In this work we employ an integrated approach involving oceanographic particle modelling, host genomics, and 16S rRNA gene amplicon (microbiome) sequencing to study the microbiome dynamics of two rafting macroalgae, *Durvillaea poha* and *D. antarctica*, enabling testing of the Intermediate Stochasticity Hypothesis (ISH; ref. 14) and Anna-Karenina Principles (AKPs; ref. 38). These two hypotheses are complementary explanations for how microbiome structure can change in response to varying environments and dysbiosis, respectively. The ISH is a modification of the Intermediate Disturbance Hypothesis[39] and suggests that alpha diversity will peak when the variability or predictability of an environment is at an intermediate point, where changes in environmental conditions lead to temporal niche differentiation−and enable co-existence of more taxa. When predictability is extremely high, competition between taxa within a community drives alpha diversity down, and when predictability is too low the environment changes too rapidly for most taxa to establish within the community. The second, complementary, hypothesis is the concept of Anna-Karenina Principles[38]−to paraphrase Leo Tolstoy: 'All happy microbiomes are alike, each unhappy microbiome is unhappy in its own way'. This hypothesis is used frequently within the explicit context of dysbiosis of microbial communities[39–42], where shifts in community structure in response to disturbances lead to a net increase in the beta diversity of a community. These two hypotheses are particularly useful for understanding host-dispersal impacts on the microbiome, because environmental variability may act similarly to a disturbance in inducing dysbiosis. We show that *Durvillaea* associated microbial communities are strongly influenced by the rafting process, exhibiting evidence for both the ISH and AKPs.

## Results
### Host genomics
Using Genotyping-by-Sequencing, we were able to assign rafts collected from the Munida transect (Fig. 1) to a range of populations around New Zealand based on the 37,012 SNPs which were retained after filtering. Based on these data, we assigned 9 rafts to populations with high confidence, 16 rafts to localized geographic regions (e.g., the Catlins, or the Otago Peninsula, Fig. 1), 10 to broader geographic regions, and 2 rafts remained unassigned. Broadly, our rafts originated from southern New Zealand, predominantly from 200 km of coastline local to the transect used for raft collection, but with some rafts having dispersed from further afield such as the sub-Antarctic Snares Islands.

### Particle modelling
Using oceanographic modelling of 99,999 simulated particles for each raft, we found that between 0.07% and 65.59% (depending on specific raft) reached the source location inferred via genetic analyses through backward advection, with a median rafting time of between 3.33 and 132.00 days. Particles that did reach their source took between 0.58 and 728.71 days to arrive. Rafts which had high confidence source assignments and/or more distant sources (e.g., Snares Islands) tended to have fewer particles reaching the source location.

We identified 21 real-world undrogued SVP (Surface Velocity Program) drifters that passed through the Munida Transect between 2015 and 2022. Variability in sea surface temperature (σ-SST) values between real-world drifters and equivalent modelled trajectories were statistically equivalent at an effect size of 0.25 ($t = 2.57$, d.f. = 20, $p = 0.009$, mean difference of −0.14, 90% confidence intervals of −0.21 and 0.07) (Fig. 2A). Comparisons of 'static' particles (i.e., kelp that remain in place, rather than undertaking a drifting process) to 'raft' particles revealed that the latter had significantly higher variation in sea surface temperature (Fig. 2B).

### Integration/synthesis
From microbial samples derived from seawater and both rafts and non-raft macroalgae, a total of 28,394 ASVs (amplicon sequence variant) were identified, and following rarefaction 11,764 ASVs remained. Rarefaction curves of both ASV count and Shannon diversity supported the rarefaction level of 4000 reads (Supplementary. Fig. 1). Raft microbiomes were distinct from both seawater and non-rafts, with increased Bray-Curtis dissimilarity observed in rafts relative to non-rafts and raft microbiomes becoming increasingly dissimilar to non-raft microbiomes with increasing rafting time. Rafts had significantly higher average dissimilarity than non-raft microbiomes ($P = 3.4 \times 10^{-11}$, $Z = −6.628$, Wilcoxon ranked sum test, Fig. 3B). The relationship between dissimilarity and raft time, sea surface temperature, and standard deviation of SST suggested that for beta-diversity, raft time and standard deviation were most important, with less importance attributed to sea surface temperature (Table 1; Fig. 3c−e), these models explain 11% of deviance in Bray-Curtis dissimilarity, with standard deviation contributing more when higher. There was no evidence of

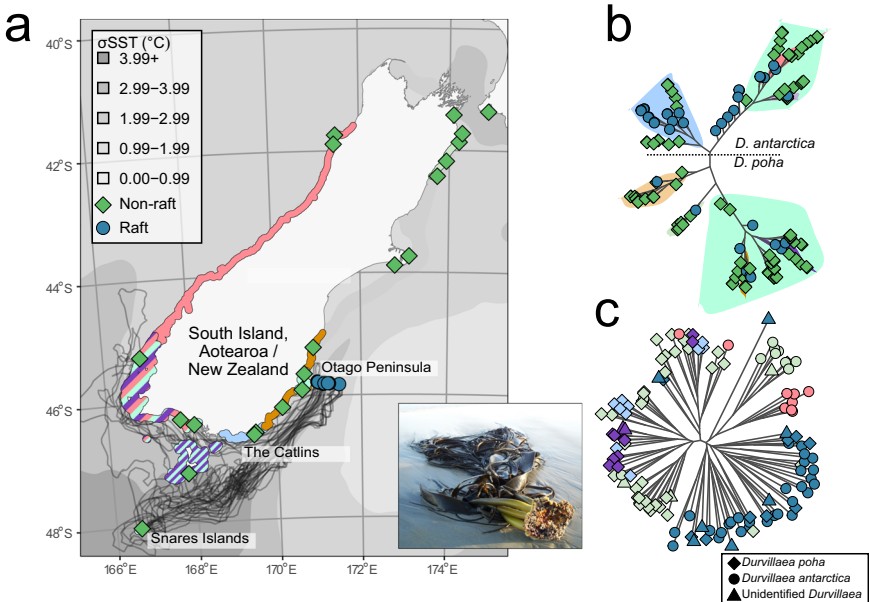

**Fig. 1 | Map of sampling locations with associated rafting trajectories and microbial and host relationships. a** Map of sampling locations for source population genetic data and raft genetic samples. Coloured areas around the coast represent 'source' regions from which rafts could have originated, with multi-colour zones indicating overlapping regions with differing levels of phylogenomic uncertainty (e.g., all regions with pink represent a single region from which some rafts originated). Background shading of the ocean represents standard deviation in sea surface temperature over 5 years. Black lines indicate successful OpenDrift trajectories for three example rafts; not all trajectories can be shown on this plot. **b** phylogenetic relationships of kelp rafts and non-rafting kelp, highlighted backgrounds represent source population locality. **c** Neighbour-joining tree of microbiome samples of rafts and non-rafts, coloured shapes indicating source zone in (**a**) based on merged replicates of microbiome samples. Species of *Durvillaea* are indicated by shape.

kelp species-specific (*D. antarctica* vs *D. poha*) patterns observed via clustering, with non-raft microbiomes clustering together regardless of host species (Figs. 1c and 3a).

As with beta-diversity, rafts tended to have higher alpha diversity (Fig. 4; Table 2), which exhibited a hump-shaped relationship with σ-SST. The final GAMs (generalized additive models) used to predict species richness/evenness consisted of sea surface temperature and its standard deviation, and raft time for evenness (Table 2). Raft microbiomes typically had higher ASV richness and evenness than non-rafts (mean richness: 104.4 ASVs vs 148.9 ASVs; mean Pielou's evenness: 0.63 vs 0.70).

After classifying ASVs as either core, abundant, or rare based on average abundance across raft or non-raft samples, we examined the relative contributions of these groupings to community structure in rafts and non-rafts. Core microbiomes were defined based solely on non-rafts, as we were interested in seeing how the dominance of a 'normal' core microbial community is affected by rafting, 14 ASVs were found as core across non-rafts (Supplementary. Fig. 2). We found that within rafts, microbial communities were dominated by abundant microbes (55%), and similar contributions of core and rare microbes (mean percentages of 25% and 20%). Conversely, non-rafts had much higher contributions of core microbes (59%), and lower contributions of abundant and rare microbes (31% and 10% respectively) (Fig. 5).

iCAMP analyses of ecological processes shaping microbiome assembly revealed that non-rafts were primarily shaped by homogeneous selection and ecological drift. Conversely, rafts were less driven by selection (30.8% vs 65.4%), with a much larger contribution of dispersal limitation (36.4% in rafts vs 7.8% in non-rafts; Fig. 6b; Supplementary Table 1). Although the primary difference across all phylogenetic bins was a reduction in homogeneous selection and an increase in dispersal limitation, we note that one phylogenetic bin was composed almost entirely of ASVs assigned to the genus *Granulosicoccus*, a widespread macroalgae-associated taxon, and an identified core taxon. This *Granulosicoccus* bin was dominated by homogeneous selection in non-rafts (82.1%) but not in rafts (9.1%) (Supplementary Table 2). These results were reinforced by our observation that rafts were typically in a state of dysbiosis, with dysbiosis scores ranging from −0.2 to 0.35, while non-rafts ranged from −0.28 to 0.07; specific score was significantly associated with standard deviation of SST (Fig. 6b). Finally, we noted that raft communities had significantly higher weighted mean rRNA operon counts than non-rafts (Supplementary Fig. 3).

## Discussion
Our findings indicate that changes in microbiome structure during long-distance movement are predominantly influenced by the variability of the environment that hosts traverse (Figs. 3, 4). This influence follows the relationship expected under the Intermediate Stochasticity Hypothesis (ISH), which posits that environmental predictability can contribute to the relative strengths of deterministic and stochastic processes shaping the community, leading to changes in alpha diversity[14]. Microbiome richness in rafts is better explained by a combination of raft status (raft or non-raft) and standard deviation of sea surface temperature (σ-SST), rather than rafting time (Fig. 4/Table 2). This relationship between species richness and σ-SST (Fig. 4) is non-linear and peaks at intermediate levels of variability, aligning with the changes in alpha diversity predicted by the ISH[14].

The variation in sea surface temperature found in many of our rafts may also be partly explained by many rafts having crossed multiple water masses, as our sampling transect crosses neritic/coastal, subtropical, and sub-Antarctic water masses. Transitioning through these water masses would not only increase the range of SST to which a raft is exposed but would also alter the exposure to macro- and micro-nutrient poor/rich waters—perhaps promoting increased variation in the microbiome. Therefore, although we focus principally on sea surface temperature and its variability, we do not suggest that it is sea-surface temperature that inherently drives variation in the microbiome. Instead, we suggest that mean SST and SST variability are useful

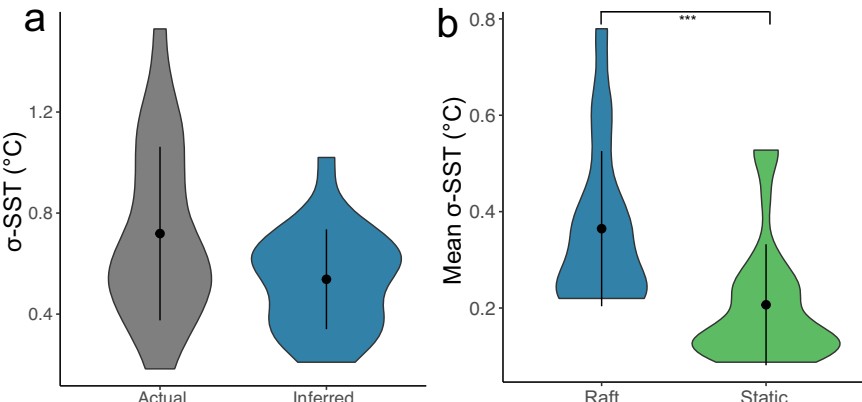

**Fig. 2 | Comparisons of sea surface temperature variability in modelled and empirical data.** Comparison plots of sea surface temperature (SST) along modelled OpenDrift Trajectories and empirical drifter data, **a** standard deviation for a real-world drifter, or mean standard deviations for modelled trajectories from the same timeframe as the real-world drifter, distributions were equivalent based on two one-sided *t*-tests ($t = 2.57$, d.f. = 20, $p = 0.009$, mean difference of −0.14, 90% confidence intervals of −0.21 and 0.07). **b** The standard deviation of SST for modelled kelp rafting trajectories, and the standard deviation of SST over the same time period at the inferred source location of the raft, differences were significant based on a Welch−two-sided *t*-test ($t = 4.51$, $p = 2.9 \times 10^{-5}$, $n = 34$ for both groups). Whisker plots within violin plots indicate median and interquartile range.

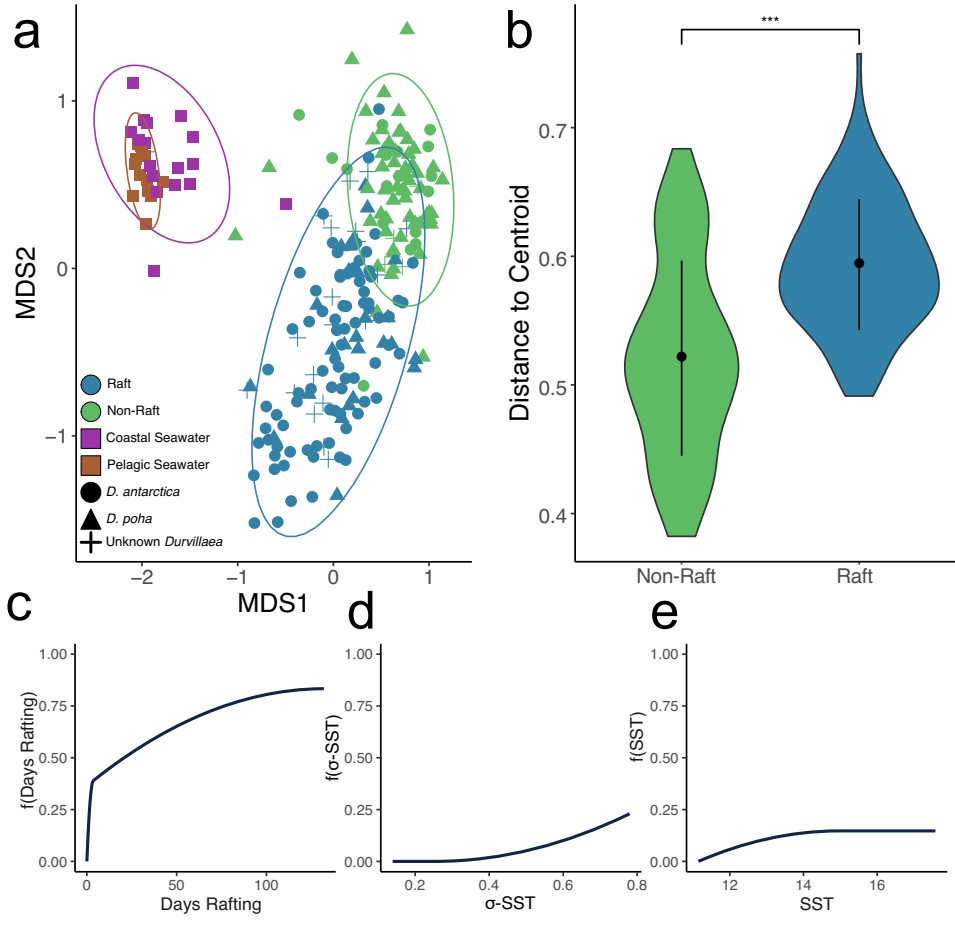

**Fig. 3 | Influence of host and environmental characteristics of microbiome beta-diversity of *Durvillaea*. a** nMDS ordination of microbiomes associated with kelp rafts, non-rafting kelp, and seawater. Shape indicates sample type (algal species or seawater) and colour indicates source of sample (raft or non-raft), stress for this nMDS is 0.13, $k = 3$. **b** Distance to centroids is based on Bray-Curtis dissimilarity for microbiomes of rafts and non-rafts, asterisks indicate significance ($Z = −6.628$, $p = 3.4 \times 10^{-11}$) based on a Wilcoxon ranked sum test, $n = 81$ and 127 for non-raft and raft respectively. **c**−**e** represent the I-splines generated by the Generalized Dissimilarity Models for raft time (**c**), standard deviation of SST (**d**) and for the mean SST (**e**). Heights indicate the relative contributions of each variable to total beta-diversity, at each point. Whisker plots within violin plots indicate median and interquartile range.

proxies for a broader suite of environmental variables that interact to shape microbiome structure. Nevertheless, SST likely also directly affects raft microbiomes as temperature underlies a range of other processes such as biochemical rates[43], where colder temperatures may slow processes such as decomposition[44]. Furthermore, strong associations between SST and salinity are well described across our sampling transect[45], and in turn, salinity is associated with phosphate, nitrate, and chlorophyll concentrations[46], supporting our use of SST as

a generic proxy for other environmental conditions. Sea-surface temperature clearly delimits water masses, as shown in Fig. 1, and these water masses harbour distinct microbial communities. Thus, in addition to passing through varying temperatures, rafts may pass through distinct microbial environments—leading to exposure to an increased suite of potential colonizers.

We suggest that higher or lower levels of environmental variability result in selection for specific groups of taxa, while intermediate variability results in weaker selection. These inferences are supported by the larger contribution of stochastic processes to microbiome assembly in rafts vs non-rafts (Fig. 6a), and the increased dysbiosis scores (Fig. 6b). Previous epidemiological modelling has revealed that less environmental predictability can lead to increased invasion of opportunistic pathogens[47]. The suggested mechanism that promotes increased pathogen proliferation is that lower predictability leads to transient periods of low competition which leads to 'easier' invasion of communities—within the microbiome, this pattern would appear as increased dominance of neutral processes.

The dominance of homogenizing selection to microbiome composition in non-rafts (Fig. 6b) suggests a relatively tightly regulated microbiome in stable established ecosystems[15,40]. However, microbiomes of rafting kelp contrasted strongly with those of non-rafts, having both a reduced influence of selection on community structure

### Table 1 | Generalized dissimilarity model between beta diversity and environmental characteristics

| | Deviance explained by variable alone (%) | Variable Importance | P-value |
|---|---|---|---|
| Days spent rafting | 7.11% | 63.04% | <0.001 |
| SST | 0.11% | 0.95% | <0.001 |
| σ-SST | 0.42% | 3.7% | <0.001 |
| Total deviance explained | 11.28% | – | |

Results from generalized dissimilarity model for relationships between beta-diversity (Bray-Curtis) and sea surface temperature, standard deviation, and rafting time. Variable importance and p-values are based on 1000 permutations. Models were produced using the *gdm* R package.

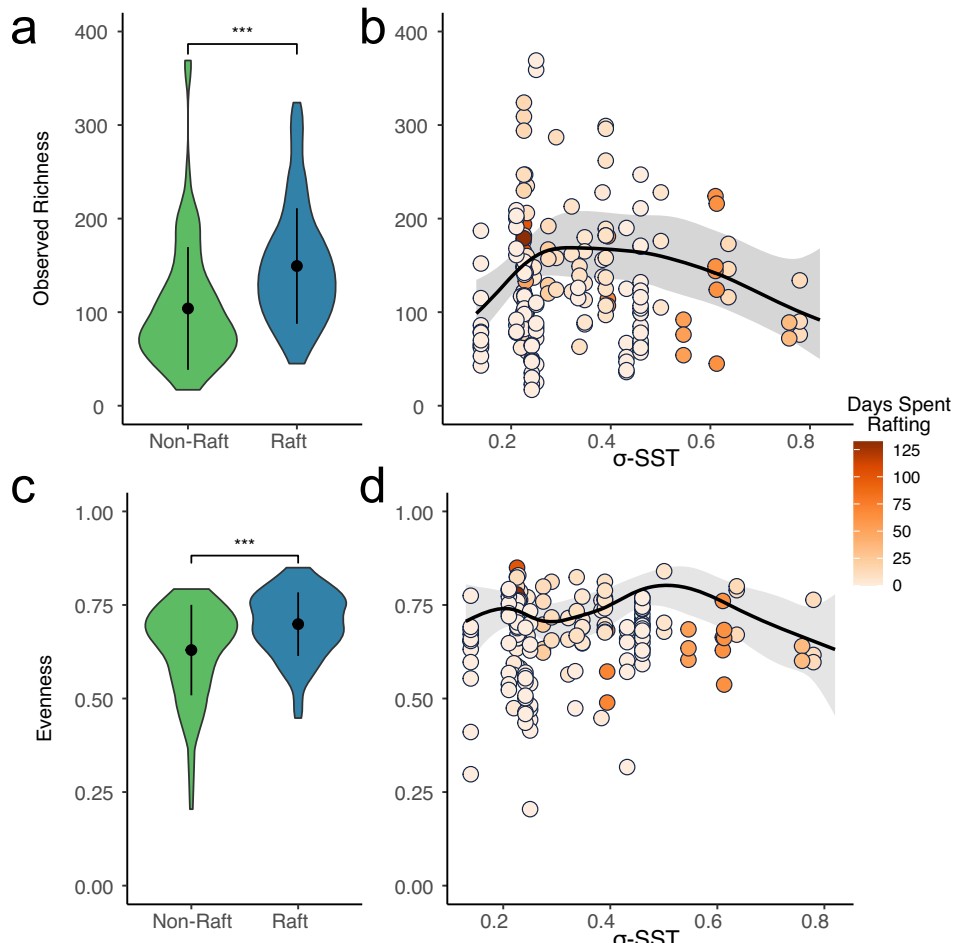

**Fig. 4 | Relationships of microbial diversity with time spent rafting and variation in sea surface temperature.** Differences in richness (**a**, **b**) and evenness (**c**, **d**) of microbiome diversity between rafts and non-rafts. **b**, **d** demonstrate the relationship between richness/evenness with the standard deviation in sea surface temperature (x-axis) and the amount of time spent rafting (shading). The shaded ribbon represents the final GAM model's 95% confidence interval. Asterisks indicate significance based on a Wilcoxon ranked sum test ($Z = -5.36/p = 8.31 \times 10^{-8}$ and $Z = -3.88/1.1 \times 10^{-4}$ for (**a**, **c**) respectively), the black line indicates the predicted values based on the final GAMs. Whisker plots within violin plots indicate median and interquartile range. These results are based on 81 non-raft and 97 raft microbial community samples.

**Table 2 | Generalized Additive Models for alpha diversity and environmental conditions**

| Component | Term | Estimate | Std Error | t-value | p-value |
|---|---|---|---|---|---|
| A. parametric coefficients | (Intercept) - Richness | 4.831 | 0.035 | 137.197 | <0.0001 |
| Component | Term | edf | Ref. df | F-value | p-value |
| B. smooth terms | s(Mean σ-SST) | 3.305 | 3.739 | 12.961 | 0.0068 |
|  | s(Mean SST) | 3.078 | 3.534 | 35.091 | <0.0001 |
| Adjusted R-squared: 0.152, Deviance explained 0.194 -REML: 1619.032, Scale est: 1.000, N: 178 | | | | | |
| Component | Term | Estimate | Std Error | t-value | p-value |
| A. parametric coefficients | (Intercept) - Evenness | 0.696 | 0.030 | 23.033 | 0.0000 |
| Component | Term | edf | Ref. df | F-value | p-value |
| B. smooth terms | s(Raft Time) | 5.398 | 6.329 | 19.560 | 0.0037 |
|  | s(Mean σ-SST) | 5.325 | 6.200 | 20.353 | 0.0025 |
|  | s(Mean SST) | 2.729 | 3.258 | 6.814 | 0.0942 |
| Adjusted R-squared: 0.261, Deviance explained 0.324 -REML: −277.821, Scale est: 1.000, N: 178 | | | | | |

Results from generalized additive models for predicting richness and evenness of microbial communities associated with *Durvillaea*. Richness was modelled using negative binomial family, and evenness with a beta regression family. In both instances, final models were chosen based on concurvity values and AIC scores. GAMs were produced using the *mgcv* R package.

and higher beta-diversity of microbes than non-rafts (Fig. 6). Host regulation of the microbiome may also be disrupted by dispersal, leading to the loss of core microbes (Fig. 5). The reduced influence of host selection and the transition towards dominance of neutral processes in community assembly (Fig. 6a) may result in a further shift of the microbiome away from a typical non-raft community and towards potential dysbiosis.

The contrast between non-raft microbiomes and the increased microbiome dysbiosis observed in rafts (Fig. 6) is a definitive realisation of Anna-Karenina Principles[38]. These principles have been applied within a range of microbiome contexts[38], with suggestions that decomposition-associated or dysbiotic microbiomes can be examined through the AKPs. Dysbiotic ('unhappy') microbiomes are thought to shift away from being dominated by deterministic processes, and instead become more shaped by stochastic processes, leading to increased beta-diversity[40]—both expectations are supported for macroalgal rafts (Figs. 3, 6). The apparent physical degradation symptoms observed on some rafts are similar to what has been noted for another brown alga, *Ecklonia radiata*, resulting from elevated temperature and decreased pH[48]. In both *E. radiata* and *Macrocystis pyrifera* these variables are also associated with microbiome dysbiosis[48,49], further supporting our hypothesis of dysbiosis in rafting algae. This raises the question of how microbiome dysbiosis affects the establishment prospects of raft offspring following dispersal to new habitat, as rafting is the principal means by which new populations of *Durvillaea* are established[21,23,37]. Rafting kelp themselves cannot re-attach to the substrate but they may still reproduce within the new environment; coalescent holdfasts of multiple individuals have been implicated in the establishment of new populations arising from even a single raft[50,51]. Interestingly, of the rafts studied here, those with the longest inferred rafting periods were in a similar or lower state of dysbiosis than rafts with a shorter inferred rafting period (Fig. 6b), and these longer-lived rafts showed surprisingly high taxonomic overlap with non-rafts (Supplementary Fig. 4), raising the possibility that a healthier microbiome facilitates longer-term dispersal. Alternatively, it may be that those rafts in better 'health' or condition are those which are capable of long-distance dispersal and thus also have a more 'typical' microbiome, especially given the link between microbiome composition and macroalgal condition[48,52].

Increased dysbiosis in macroalgal microbiomes is probably, in part, driven by disruption of host selective processes (Fig. 6). Host-imposed selection is an established driver in many plant microbial communities[40,53,54], and indeed disruption to this process has been suggested as a causative factor in microbiome dysbiosis[40].

Furthermore, the reduced influence of selection on both the whole microbiota and also, more specifically, the dominant microbial core taxon *Granulosicoccus* in rafts suggests that raft macroalgal microbiomes are in a state of dysbiosis. Thus, although a positive relationship is observed between σ-SST and dysbiosis score this may also be explained by, among other options, loss of host-regulatory capacity, rafting duration, and other unsampled environmental conditions—or more likely, a combination of all these and other interacting factors.

Despite shifts towards dysbiosis, kelp rafts are known to disperse long distances while remaining reproductively viable[21]—which raises the question of how algal rafts remain somewhat healthy despite a dysbiotic microbiome that appears geared towards ill health. However, this may be explained by recent work which shows floating *Durvillaea* is able to maintain high antioxidant and phlorotannin concentrations, despite sub-optimal conditions, and these activities may limit the extent of bacterial-induced decay and facilitate long-term rafting, alongside preservative effects of cold oceanic conditions[55]. Similar results have been observed in rafts of both *Macrocystis pyrifera*[56] and *Sargassum spinuligerum*[57], indicating that buoyant macroalgae may have physiological defences that inhibit degradation in the rafting environment. Photosynthetic capability in rafts of both *Macrocystis*[56] and *Durvillaea*[55] has been demonstrated to be reduced relative to non-rafts, suggesting that continued, albeit reduced, photosynthesis in the rafting environment counteracts degradation processes during the raft period.

Macroalgal rafts of *Durvillaea* are known to disperse many thousands of kilometres, sometimes washing up on distant shores after years at sea[21], and the rafts tested in this study reveal only the initial stages (up to 132 days) of this rafting process. This raises the question of whether, over longer voyages, hosts retain some capacity to maintain the microbiota in the face of extremely changeable environmental and host conditions. Indeed, biofilm experiments have revealed that turnover dictates community succession rather than priority effects[58]; if this is the case for host-associated biofilms, then the successional changes prompted by rafting may overwhelm any priority effects. Our data strongly suggest that rafting triggers major successional changes in community assembly, due to the increased abundance of faster-growing bacteria relative to non-rafts (Supplementary Fig. 3)[59]. We thus suggest that the likelihood of joint host-microbe dispersal over broader scales is likely linked to the maintenance of selective pressures regulating the community.

Highly dynamic environments, such as those experienced during long-distance ecological movements, have major impacts on microbial community assembly and structure. If such extreme community

change is ubiquitous for stochastic long-distance movements, we may find that many organisms are unable to bring their original microbial community with them during range shifts and species invasions—potentially resulting in selection for hosts which have flexibility with regards to their microbiome. Indeed, recent work has shown that invasive lineages of one alga are more flexible with regard to microbiome composition[60]. Selection for microbiome flexibility could facilitate future ecological invasions and range shifts.

## Methods
### Sample collection
All samples were collected under Ministry of Primary Industries Special Permits 824-2 and 644; no ethics or additional permits were required to collect these samples. Samples were collected between January 2021 and March 2022, with kelp rafts sampled opportunistically at sea along a 60 km oceanic transect (the Munida Time Series Transect, Fig. 1; Supplementary Table 3) extending east from the Otago Peninsula, southern New Zealand[61], with a total of 37 independent rafts being

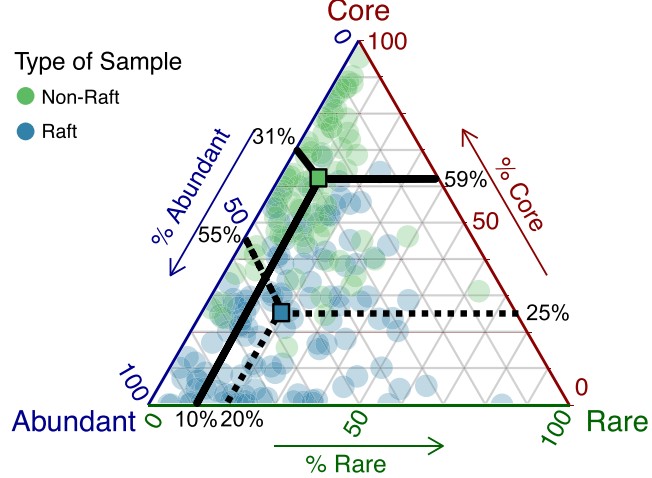

**Fig. 5 | Ternary plot of core/rare/abundant microbes for raft and non-raft microbial samples.** Ternary plot of the distributions of microbial communities associated with rafting and non-rafting *Durvillaea*. Core microbes were defined based on contributions of towards overall community beta diversity, and abundant microbes were defined as those with a mean abundance in non-rafts ≥0.1%. Coloured squares indicate means of each group, with lines indicating contributions of each community class.

collected. When rafts with coalescent holdfasts were collected (i.e., multiple kelp individuals joined at the holdfast), only a single individual from that raft was sampled. Rafts were collected via a grappling hook and brought on board the RV *Polaris II*. Microbial samples were collected from two separate blades and the palmate meristem region of the thallus; tissue was first rinsed gently with sterile artificial seawater to remove transient microbes. The palmate meristem is the section of the thallus immediately above the stipe, from which blades extend; see (ref. 62; Supplementary. Fig. 5) for a diagram. Taking care to sample tissues which had not come in contact with hands, grappling hook, or the boat, a 25 cm² area of tissue was swabbed back-and-forth with a Qiagen OmniSwab, which was then ejected into a tube of sterile DESS (20% DMSO, 250 mM pH 8 EDTA, saturated with NaCl). A similarly sized tissue piece was collected and stored in silica gel for DNA extraction. Seawater, algal microbiome and tissue samples from non-rafts were collected from *Durvillaea* populations (average of 10 hosts sampled per population) around the South Island of New Zealand using a similar procedure, to provide a reference dataset of *Durvillaea* microbiomes. In addition to biofilm samples, we also collected 2 L seawater samples at the locations where rafts were collected; these were filtered using a bleach-sterilized vacuum filter with a 0.22 μM polycarbonate filter and were subsequently stored in DESS and frozen.

### Microbiome methods
DNA was extracted from swabs and seawater filters using either the Qiagen PowerSoil or the PowerSoil Pro kit, following the manufacturer's instructions. Optional incubation steps were included, and samples were bead-beaten for 10 minutes at 25 Hz in a Domel Mill Mix to aid lysis. DNA was eluted using a two-step elution process of 50 μL of nuclease-free water at each step. 50 μL of eluted DNA was then transferred to a 96-well microtiter plate and dried using an Eppendorf SpeedVac.

16S rRNA gene library preparation was carried out on microbial DNA at Argonne National Laboratories (ANL), following standard Earth Microbiome Project sequencing protocols[16]. In short, DNA was amplified using updated 515 F (5′: GTGYCAGCMGCCGCGGTAA) and 806 R (5′: GGACTACNVGGGTWTCTAAT) primer pair to amplify the V4 region of the 16S rRNA gene. Amplicons were then pooled equimolarly at ANL and sequenced on an Illumina MiSeq. Amplicon sequences were first demultiplexed using IDEMP and then processed in DADA2 (v. 1.26)[63], removing PhiX and chimeric reads, and truncating reads at the first instance of a base with a quality value of less than two. Following inference of amplicon sequence variants (ASVs), ASVs were processed in the *decontam* R package (v. 1.16)[64] to remove contaminant ASVs

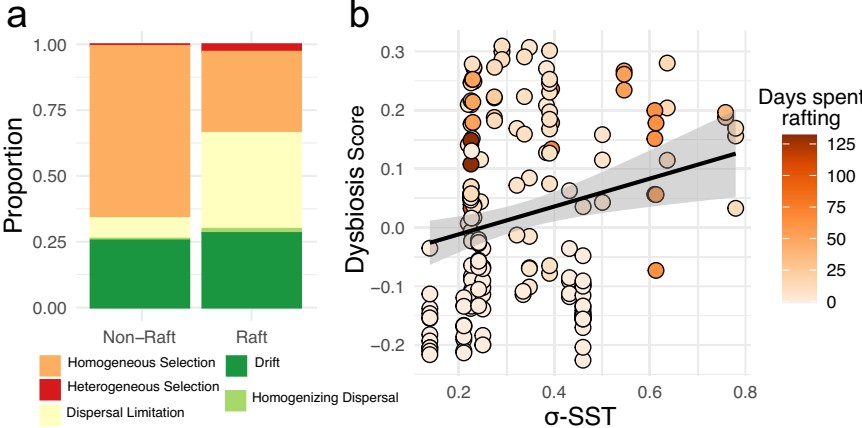

**Fig. 6 | Influence of rafting of selective pressures and dysbiosis in microbiomes. a** Contributions of different ecological factors to community assembly in non-rafts and rafts. **b** Dysbiosis score in relation to variation in sea surface temperature (x-axis) and rafting time (shading), values above 0 indicate dysbiosis, those below 0 indicate a 'normal' microbiome. The line represents linear regression, with the ribbon indicating the 0.95 confidence interval.

using a prevalence approach (removing ASVs more abundant in negative controls than in samples), using a threshold of 0.1. Finally, amplicon sequence counts were rarefied to 4000 reads per sample (Supplementary. Fig. 1) using the *rarefy_even_depth* function in the *phyloseq* R package (v. 1.4)[65]. Bioinformatic scripts for these analyses are available at https://github.com/wpearman1996/kelp_rafting_MS. Phylogenetic inference of relationships between ASVs was conducted using FastTree (v 2.1.11) with a GTR-CAT model, sequences were aligned using MAFFT v7.505 with FFT-NS-2 and max iterations of 0. Taxonomic classification of ASVs was performed using lotus2[66] with the taxOnly option, using BLAST alignment against the PR2[67], SILVA[68], GreenGenes[69], and HITdb[70], with preferential treatment of databases in the respective order (i.e., PR2 highest priority, and HITdb lowest priority).

For each microbial sample (raft and non-raft), we calculated species richness, Pielou's evenness, and Bray-Curtis dissimilarity between samples. To demonstrate the microbiome dissimilarity amongst these populations (Fig. 1C), we merged replicate blade microbiome samples for each individual using the merge_samples function in *phyloseq* and created a neighbour-joining tree using *ape* (v. 5.7-1) based on Bray-Curtis dissimilarity of models.

To understand how community structure varied broadly across raft and non-raft communities, we conducted analyses based on three classifications of ASVs−core, abundant, and rare. We did this separately for rafts and non-rafts, classifying ASVs as either core, abundant (non-core microbes with a mean abundance >0.1%[71], and rare (non-core microbes with a mean abundance ≤0.1%). Core microbiomes were determined with non-rafts which originated from the same geographic regions as the rafts using the spatial method from ref. 72 with the last 3% decrease in explained Bray-Curtis dissimilarity. This approach first characterises the explanatory value of each microbe to overall beta-diversity (based on Bray-Curtis dissimilarity) and then identifies the core taxa as those which result in a 3% or more decrease in Bray-Curtis dissimilarity (See[72] for an in-depth explanation). Although not based solely on occupancy, this method necessitates microbes must be found at least 80% of samples to be classified as core. Ternary plots were created with the *GGTern* R package (v. 3.5.0)[73].

Relative contributions of different ecological processes to community structure were determined with the iCAMP (v. 1.6.3) framework using the *icamp.cm* function with bin sizes of 24, and the confidence method of testing[15,74]. This method broadly estimates the contributions of selection, drift, and dispersal to community structure based on measures of pairwise taxonomic and phylogenetic dispersion. Beta nearest taxon index was used as a metric of phylogenetic turnover, for non-raft samples phylogenetic turnover was calculated within populations to account for meta-community structure. In order to assess the influence of growth rate on community assembly, we exploited the known relationship between rRNA operon copy number and maximum growth rate[75] by aggregating microbiome data to the genus level and retrieving the mean rRNA operon count for each genus from rrnDB (v 5.8)[76]. A community level rRNA trait was then calculated as the weighted mean of the genus level abundances of rRNA operon counts following[59].

## Host genomics

For host genomic analysis, DNA from kelp tissue was extracted following an in-house protocol (developed to maximise DNA quantity and purity for problematic kelp extractions), in which dried tissue was first ground to a fine powder in 20-second bursts at 25 Hz with 4 mm steel balls in a Domel Mill Mix. To approximately 100 mg of powder, 1.2 mLs of lysis buffer (6 M Guanidine Hydrochloride, 250 mM EDTA, 100 mM Tris-HCl, 1% Sodium metabisulphite, 1% polyvinylpyrrollidone 30 K), and 20 μL of Proteinase K (20 mg/mL) was added, and mixed thoroughly, followed by overnight incubation at 65 °C. The lysate was

centrifuged for 2 minutes at 10,000 × g and 1 mL of supernatant was transferred to a tube containing 400 μL of 5 M potassium acetate. This reaction was thoroughly mixed and incubated for 20 min on ice, followed by another 2 minutes of centrifugation at 10,000 × g. 1 mL of the supernatant was transferred to a tube of 1 mL of precipitation solution (20% PEG8000, 1.2 M NaCl), and mixed thoroughly, followed by another 20 min incubation on ice. The mixture was centrifuged for 10 × min at 15,000 g, and the supernatant was discarded. The DNA pellet was then washed twice with wash solution (70% ethanol, 30% TE buffer), and resuspended in nuclease-free water. Finally, the DNA solution was cleaned using the Qiagen PowerClean Pro kit, following the manufacturer's instructions.

Genotyping-by-Sequencing (GBS) host libraries were prepared following a modified protocol from Elshire et al.[77] that has been optimized for work with *D. antarctica*[35]. 750 ng of DNA from each sample was digested using the *PstI* restriction enzyme, DNA was suspended in 17.8 μl of nuclease-free water, and 0.2 μL of *PstI*-HF (NEB #R3140L) and 2 μL of 10X CutSmart buffer, alongside 2.25 ng of PstI sequencing adaptor. Reactions were mixed briefly by pipetting and were then spun down and incubated at 37 °C for 2 h. Following digestion, 5 μL of 10X T4 DNA Ligase reaction buffer, 1 μL of T4 DNA ligase (NEB # M0202LVIAL), and 24 μL of nuclease-free water were added to each reaction, mixed by pipetting, and then incubated using a PCR program of 16 °C for 30 min, 37 °C for 2 min, 16 °C for 30 min, 37 °C for 2 min, 16 °C for 30 min, 80 °C for 30 min, and then storage at 4 °C.

The digested and barcoded DNA was then cleaned using a Qiagen MinElute 96-well plate on a vacuum plate manifold using a vacuum pressure of 800 mbar was applied, and the wells were washed twice with 30 μL of nuclease-free water. Finally, DNA was resuspended in 23 μl of nuclease-free water, by brief incubation at room temperature for 5 minutes, followed by 30 cycles of pipetting. A PCR reaction consisting of 10 μL of the cleaned DNA, with 25 μL of 2X MyTaq HS Master Mix, and 1 μl of forward and reverse primer was conducted. The PCR protocol consisted of 72 °C for 5 min, 95 °C for 1 min, then 24 cycles of 95 °C for 30 s, 65 °C for 30 s, 72 °C for 30 s, followed by a 5 min hold for 4 °C.

PCR success was assessed visually based on gel electrophoresis, and samples were pooled roughly based on the gel fluorescence of the PCR products. DNA libraries were size selected using gel excision to retain DNA between 200 and 600 bp and then sequenced at the Biomolecular Resources Facility at the Australian National University on an Illumina HiSeq with 75 bp paired-end reads.

The GBS sequence library was first quality-filtered using FastQC[78] and trimmed to remove adaptors, PhiX reads, and homopolymers. Libraries were then demultiplexed using STACKS[79], using *process_radtags* with the PstI enzyme, and inline barcodes. Following established protocols and including those samples from refs. 21,80, we aligned demultiplexed reads to a *D. antarctica* reference genome using BWA (v. 0.7.17)[81] and then processed into SNPs using the STACKS *ref_map* and *populations* module, with values of -p 1, -r 0.2. We then removed SNPs with a minimum depth of <5, Minor Allele Count (MAC) <2, and missingness >0.2, finally, we removed individuals with fewer than 5000 SNPs per sample.

Following bioinformatic filtering, we conducted phylogenetic inference using IQ-Tree (v. 2.2)[82]. Samples were assigned to populations based on bootstrap (n = 1000 bootstraps) results and proximity to other known samples. Initial analyses indicated all rafts originated from within New Zealand's Exclusive Economic Zone, and thus we did not include GBS samples from global datasets. Samples were assigned to clades based on bootstrap values >80, i.e. assigned to a regional group based on their position within clades containing samples of known origin. Origin regions for particle modelling for each raft were based on the maximum geographic range from which samples originated within each clade. Specifically, we traversed up the tree until a node with >80 support was found, then the geographic range between

the two most distantly found locations within that clade was used as the inferred source location.

## Particle modelling

To model potential backward trajectories of each collected kelp raft, we used a Lagrangian drift particle model executed using the Leeway module of OpenDrift (v 1.9.0, https://opendrift.github.io[83], a widely used oceanographic software package that has been validated against real-world data[84,85] and used to model, among other things, eDNA[86] and marine debris[87] transport. At each raft collection site (Fig. 1), 99,999 particles were randomly seeded laterally within a 1 km radius of the collection coordinates (Supplementary Table 3) and at times ±1 day of the collection date[86]. These particles were then advected backward in time using an hourly time-step (i.e. starting at the collection site and tracking backward to potential raft origin locations). An hourly time step was chosen so that particles were unlikely to move more than one grid cell per time step. Backward trajectories were subsequently compared to simulations with 15, 30, and 120 min time-steps to test if the Lagrangian simulation had converged to a stable result—changes in time-step size did not result in any qualitative differences in our results (see Supplementary. Figs. 6–7 for comparison of time-steps). Using an Euler propagation scheme, advection occurred as a function of ambient water current (Global Ocean Reanalysis; GLORYS12; 10.48670/moi-00021) and wind[88] vectors, which varied depending on the drift properties of each seeded particle. Both horizontal and vertical diffusivity coefficients were set at 0 m s⁻¹ (i.e., no additional diffusion). Sensitivity tests conducted across a plausible range of horizontal diffusivity values[89,90] did not yield any qualitative differences in our results (see Supplementary Figs. 8–9 for sensitivity testing of diffusivity coefficients). To cover a range of potential raft geometries, seeded particles were parameterised as one of three particle types (PIW-1, PIW-5, PIW-6 as defined in the Leeway model; https://opendrift.github.io[91,92]). Each of these particle types represents the drift properties of a person-in-water (PIW) in one of three states (unknown state, scuba suit—face up, deceased—face down, for PIW-1, 5, and 6, respectively; Supplementary Table 4). In the absence of specific drift property measurements for *Durvillaea* rafts, these three particle types were chosen, based on our own qualitative field observations, as the most similar to a rafting piece of kelp with an attached holdfast, in terms of dimensions, buoyancy, and drift properties (Supplementary Table 4). Smaller *Durvillaea* rafts like the ones we collected are comparable in size and shape to a person lying horizontally on the water surface. We do, however, acknowledge that while this ensures variety in particles and resultant trajectories the parameters are somewhat arbitrary in the absence of kelp-raft-specific parameters.

Backward advection of each particle from the point of sample collection continued until one of three conditions was met; the particle either left the area of interest (20–50° S, 120–179° E), came within 2 km of the predefined coastal target zone (Fig. 1), or had been drifting for two years without leaving the area of interest or entering the target zone. The target zone for each collected raft was defined a priori based on genetically inferred source locations of the kelp (see above). Particles that came into contact with coastline that was not identified as the target zone were assumed to move offshore once current and wind conditions allowed (i.e., they were not permanently stranded or deactivated). The number of, and time taken for, particles to reach their respective target zone was calculated for each raft. Equivalent models were produced based on 9999 particles released at the site and date of the passage of a real-world drifter through the Munida Transect, although advection occurred for up to 2 years before the collection date or till the particle collided with the coast. Real-world drifters were identified from the Global Drifter Program[91] and were selected on the basis being of being undrogued. Drogues extend to 15 m depth so their presence leads to drifter dispersal being heavily shaped by

geostrophic and Ekman currents; conversely, the trajectory of undrogued drifters is more shaped by wind-drag on the exposed portion of the drifter and near surface wind-driven and wave-induced Stokes drift rather than by prevailing geostrophic and Ekman currents alone[92,93]. Because kelp rafts are known to be heavily influenced by sea surface processes via Stokes drift[23], undrogued drifters represent a more realistic analogue for a kelp raft than drogued drifters.

## Sea surface temperature data

Environmental variability can determine the extent to which deterministic and stochastic processes dominate community assembly[14]. To examine the extent to which environmental variability shapes microbial community assembly, we calculated the standard deviation of sea surface temperature (SST) along each raft trajectory, and the standard deviation of SST for the 11 days prior to non-raft kelp samples being collected. Daily SST data were retrieved from the Operational Sea Surface Temperature and Ice Analysis (OSTIA) system at a resolution of 1/20 degree[24]. We extracted SST for each time point in a raft trajectory (or in the 11 days prior to non-raft collection —the median rafting period), using the *extract* function in the R package *terra*[94]. The mean raft time, SST and standard deviation for each trajectory or sample was calculated, and the median of these values was calculated as the representative SST and standard deviation for each sample.

To validate the use of OpenDrift trajectories, we used data from the Global Drifter Program[91,95] first identifying any SVP drifters which had passed through the Munida transect between 2015 and 2022 using the 6-hourly interpolated data (https://doi.org/10.25921/7ntx-z961). Because kelp rafts are known to be heavily influenced by surface-level waves, currents and wind[23], we only examined the trajectories of drifters that had lost their drogue (i.e., drifters driven by sea surface processes rather than by prevailing currents alone[96]. For each of these drifters, we extracted the trajectories for the 11 days (11 days was the median successful macroalgal rafting period) prior to passage through the transect and calculated the standard deviation of SST as above. Equivalence testing was conducted between these data and the comparable trajectories generated for each drifter, using the mean standard deviation of SST for the 9999 comparable trajectories generated for each drifter. Equivalence testing was conducted with two one-sided *T*-tests with an effect size of interest of 0.25 as implemented in the R package *TOSTER* (v. 0.7.1)[97].

## Integration/synthesis

For dissimilarity analyses between rafts and non-rafts (i.e., average dissimilarity of a raft to a non-raft), we only compared microbiome samples from geographic areas which overlapped with potential sources of kelp rafts (i.e., we only retained non-raft microbiome samples from the Catlins, Stewart Island, and the West Coast n = 46). For these analyses, we defined 'dysbiotic' communities following[98] using the *dysbiosisR* package with the function *euclideanDistCentroids*[99]—this calculates the difference between each raft's distance to the non-raft and rafting centroids such that values greater than 0 indicate dysbiosis relative to non-rafts. For all other analyses, we utilized all non-raft and raft microbiome samples (restriction to individual raft sub-samples did not qualitatively alter our results). Bray-Curtis distances between samples were modelled using Generalized Dissimilarity Models (GDMs) in the *gdm* R package (v. 1.5.0-9.1). Generalized Additive Models (GAMs) were conducted in the R package *mgcv* (v. 1.8.41)[100], using a negative binomial family for ASV richness and beta regression family for evenness (because values are bounded between 0 and 1), using raft time, sea surface temperature, and standard deviation of sea surface temperature as smoothed terms, while using raft status (raft or non-raft) as a linear term. Variable selection for GAMs was conducted based on AIC and concurvity values.

**Reporting summary**

Further information on research design is available in the Nature Portfolio Reporting Summary linked to this article.

## Data availability

The genetic and source data generated in this study have been deposited in the Aotearoa Genomic Data Repository database with the following https://doi.org/10.57748/m4fe-0d07 and https://doi.org/10.57748/RDXN-1598. These data are available under restricted access as they arise from culturally significant species in Aotearoa New Zealand; access can be obtained by contacting the Aotearoa Genomics Data Repository, who will liaise with indigenous groups regarding data access. The remaining samples which have not been destructively sampled are housed at the Portobello Marine Laboratory, University of Otago, Dunedin, New Zealand.

## Code availability

Code underlying the results presented in this paper is available at: https://github.com/wpearman1996/kelp_rafting_MS and at https://doi.org/10.5281/zenodo.13910085.

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

## Acknowledgements

We thank Xiaoyue Liu and Frances Perry for their help collecting kelp rafts, alongside the skippers of the *Polaris II*, Bill Dickson and Mark Elder. We also thank the various members of the Munida trips who helped spot kelp rafts, and Linda Groenewegen and Doug Mackie for their help in the lab. WSP was supported by a University of Otago Doctoral Scholarship and PhD funding from Departments of Marine Science, Anatomy, and Microbiology and Immunology. CIF and GAD were supported by a Marsden Fund grant, managed by Royal Society Te Apārangi (MFP-20-UOO-173). Research costs were also funded by a Rutherford Discovery fellowship to CIF (RDF-UOO1803). We wish to acknowledge the use of New Zealand eScience Infrastructure (NeSI) high-performance computing facilities, consulting support and/or training services as part of this research. New Zealand's national facilities are provided by NeSI and funded jointly by NeSI's collaborator institutions and through the Ministry of Business, Innovation & Employment's Research Infrastructure program. URL https://www.nesi.org.nz.

## Author contributions

W.S.P., S.E.M., C.I.F. and N.J.G. conceived the study. W.S.P. and K.I.C. conducted field work, K.I.C. organized boat expeditions. W.S.P. performed lab work. G.A.D., W.S.P. and R.O.S. conducted data analysis. R.O.S. and G.A.D. conducted the oceanographic modelling. N.J.G., C.I.F. and S.E.M. provided supervisory oversight and contributed to study design and planning. W.S.P. wrote the manuscript draft with input from all authors. All authors read and approved the manuscript.

## Competing interests

The authors declare no competing interests.
