## [Peer Review file · Nature Communications]

Host dispersal relaxes selective pressures in rafting microbiomes and triggers successional changes

Corresponding Author: Dr William Pearman

Version 0:

Reviewer comments:

Reviewer #1

(Remarks to the Author)

Review of "Host dispersal relaxes selective pressures in rafting microbiomes, triggering successional changes" by Pearman et al.

This very well written manuscript describes changes in microbiome composition of seaweeds that have become detached and formed floating rafts in oceanic waters off Aotearoa, New Zealand, conducted a comprehensive comparison between rafting seaweeds and conspecifics remaining attached to the benthos in their 'native' / donor populations, and related differences to environmental variables and host genetics, using innovative analyses and the context of two community ecology hypotheses. They report an increase in disease-associated bacteria in rafting kelp and increased rates of dysbiosis in rafting microbiomes compared to non-raft kelp. These changes are correlated with time spent in raft (however this relationship appears non-linear with evidence of a 'return' to non-raft microbiome composition after longer periods spent drifting) and to a lesser extent, SST, with higher variability in SST correlating with dysbiosis and a loss of core microbiome taxa present in attached (non-raft) seaweed.

The data collected are very interesting and certainly contribute to our understanding of seaweed-associated microbiomes, providing new insights into how they might change when benthic seaweeds become detached but persist as hosts for long periods of time). The authors cite other articles that postulate raft algae such as the ones sampled in this study, could be vectors of disease and other microbes into new areas..

However, aspects of the sampling design are not clearly described in the manuscript and I am concerned that some of the conclusions are not well supported by the data and that other hypotheses or interpretations might be equally as valid. Specifically:

Time spent in raft environment: I appreciate the modelling approach used to estimate length of time individual seaweeds spent in the rafting environment and that it is difficult to obtain such data from sampling alone. However, this did appear to me to be a bit of a limitation in terms of interpreting the data presented here. Disease-associated bacteria are often also ecologically relevant as decomposers and often over-represented on older tissues, playing roles in natural senescence and microbial decomposition. A recurring question I asked while reviewing this manuscript was how you can know whether this is really disease or just natural processes? Without knowing how long the kelp have been rafting for, or their condition (e.g. visible signs of disease [these were mentioned once in the ms. but no data were presented], pigmentation, PAR, fouling, grazing, etc.), it's very difficult to separate disease and normal senescence related decomposition. Seaweed-associated microbiomes change dramatically with dramatic changes in environment and with host age and condition. Teasing these processes apart from rafting specifically, is difficult, but I appreciate the value in trying, in the context of dispersal. Similarly, it is difficult to assign causality to a positive relationship between SST (or variation therein) and dysbiosis in the microbiome (L458-460) as there are too many coincident, unexplored factors that could provide alternative explanations.

Apparent assumption about return of raft algae to "target zone": At several points in the ms. (L269 – 283 and L532) it appears that there is an assumption or expectation that after some time spent in the drift, the kelp will return to a benthic lifestyle? Perhaps I am misinterpreting the authors' meaning, but kelp do not reattach after they become detached and, whilst sometimes remaining reproductively viable for some time, will either sink into deep ocean sediments, be consumed by herbivores or become part of the detrital food web via beach casting. I question whether this apparent assumption has

influenced the authors' interpretations of their data. The interpretation that having a healthy microbiome facilitates longer-term dispersal is interesting (L535). Equally, though, microbiome 'healthiness' could simply be a function of seaweed condition, which is the result of multiple biological and physical factors.

Host genetics and microbiomes: Species-specific seaweed-associated microbiomes are typical, yet not observed here, where microbiomes were similar in two non-raft species of kelp showing no such patterns, according to 16S (L389-390; Fig. 3a). Yet later, when population-level genetics are assessed against microbiomes, there a strong relationship is reported (L439-442), with 21% and 12% of the variation in microbiomes explained by host genetics on non-raft and raft kelp respectively. I don't know how to interpret this and this apparent discrepancy is not discussed further in the ms.

Sampling design: How many replicate individuals were sampled per location along the transect? Was seawater also collected at non-raft locations? I could not see this reported in the Methods (but data appear in the Results). Which replicates were merged as per L112 and L152? Which species were sampled at which location? Although all the technical methodologies appear sound and appropriate to me, it is difficult to critically assess some of the analyses and conclusions without understanding the sampling design in more detail.

Other minor comments / questions?:

- L73 – How big is a raft of *Durvillaea*? One individual? How large an area is 70 million rafts of *Durvillaea*?
- Intermediate Stochastic Hypothesis and Ana Karenina Hypothesis should be clearly defined in Introduction for non-expert readers outside of the authors' discipline. Alternatively, remove from the manuscript as they are inadequately explored or explained in the context of the data, as currently presented.
- L94 - Diagram /further description indicating palmate meristem region of the thallus would benefit non-expert readers from outside of the authors' discipline.
- When discussing how core microbiomes were identified and classified, it would be helpful to explain the proportion of samples in which microorganisms needed to be present to be considered 'core' (in addition to the statistical methods used).
- Description of how DABs were identified and analysed is a bit confusing (L166 – L185. Perhaps the paragraphs are around the wrong way?
- L284 – is "Environmental Data" SST only? If so, consider just calling it SST?
- L289 – are "in-place kelps" the same as "non-raft kelps"? Please be consistent with labels.
- Explanation of drogue is not clear to me - if they have lost their drogue, are they driven by sea surface processes or are they driven by prevailing currents alone?
- L332-333 – what the "geographical range" is, is not clear from the text. Perhaps cite the relevant figure?
- L356-357 - So if the confidence that a particular raft came from a particular location was high, the likelihood of that raft returning to that location was lower?

Reviewer #2

(Remarks to the Author)

Below is my review of "Host dispersal relaxes selective pressures in rafting microbiomes, triggering successional changes" by Pearman et al. I was asked to focus on the particle modeling specifically, and that is the main emphasis of my review.

The manuscript describes analysis of microbiomes taken from kelp blades from two populations in waters near southern New Zealand in 2021 and 2022. One group had drifted with ocean conditions (currents, waves, and winds) and is referred to as rafted. The other samples, non-rafts, came from established populations in nearshore environments, and thus did not experience variable open ocean conditions. Ensembles of trajectories of rafts were estimated using a particle tracking model, driven by currents from ocean reanalyses and winds from an atmospheric product. Sea surface temperature (SST) was sampled along trajectories from a data product to obtain SST variability possibly experienced along the track. Differences in microbiomes and microbiome diversity were established between the populations. A model suggests that dissimilarity increased with raft duration. A model of microbiome diversity richness showed greatest richness for intermediate values of SST variability, less dependent on duration.

The paper is well-written and results are interesting to ecological and oceanographic communities. Conclusions are well founded based on results obtained. The particle tracking modeling is state-of-the-art, with appropriate, high quality input fields for currents, winds, and SST. I recommend publication with minor modifications.

Larger comments

1) The authors should include information relating to sensitivity tests done with the trajectory time-stepping. For example, had the authors chosen time-steps one half or one quarter as large as those used, would the statistics of their results change? I do not believe any information about time-stepping is presently included in the manuscript but should be. I assume that these sensitivity tests were carried out and reasonable convergence in statistics was achieved. The authors could confirm this.

2) SST variability is treated as a likely cause for stochastic forcing. The authors might speculate on whether temperature variability is likely the driving factor or if it is a proxy for mixing with different water masses with different biological or nutrient environments. I would have thought the greater influence would be mixing with different water masses with exposure to different communities than temperature variability alone. The authors might mention for example whether the SST variability

is a proxy for other factors or if, in their opinion, it is the temperature variability that is critical.

Minor comments

3) Line 11: don't -> do not

4) Line 257: "randomly seeded laterally within a 1 km radius of the collection coordinates."

5) Line 264: I request a bit more information about PIW-1, PIW-5, and PIW-6. These are available in the opendrift documentation, but I think information here can be included, specifically what PIW stands for and what the numbers indicate.

6) Line 357: "reached the source location inferred via genetic analysis through backward in time advection, with a median..."

7) Line 359: replace target with source (source is clearer in this context).

Version 1:

Reviewer comments:

Reviewer #2

(Remarks to the Author)

Below is my review of the revision to "Host dispersal relaxes selective pressures in rafting microbiomes, triggering successional changes" by Pearman et al.

The authors have done a good job of addressing my points from the first review. However, I disagree with one point and would like the authors to correct what I view is now a growing misperception.

C19) 1) The authors should include information relating to sensitivity tests done with the trajectory time-stepping. For example, had the authors chosen time-steps one half or one quarter as large as those used, would the statistics of their results change? I do not believe any information about time-stepping is presently included in the manuscript but should be. I assume that these sensitivity tests were carried out and reasonable convergence in statistics was achieved. The authors could confirm this.

R19) We now include in the supplementary data a range of sensitivity tests to both the trajectory time-stepping and the horizontal diffusivity. Although the sensitivity tests show that our results are not affected by a change in time step, we also highlight that our chosen timestep meets the Courant-Friedrichs-Lewy (CFL) stability criterion. The CFL criterion is:

$$\Delta t = (dx / V_{max}) * C$$

where dx is the model grid resolution in m, V max is the maximum velocity anticipated in a grid cell, C is a constant = 1 to meet the CFL criteria and Δt is the minimum time step necessary to meet the CFL stability criterion.

The GLORYS12 ocean model used in our manuscript has a spatial resolution(dx) of 1/12° lat-lon, which is approximately 9 km at the latitude of New Zealand. In the vicinity of southern New Zealand / Tasman Sea, velocities in GLORYS grid cells are typically < 0.5 m/s. They get slightly larger in the core of the Southland Current, inshore of which some of our sampling took place. but even then, they are < 1.0 m/s (see e.g. <https://data.marine.copernicus.eu/-/hj20k5w1bt>).

Using a conservative value of 1.0 m/s for Vmax provides a minimum Δt required to meet the CFL criterion of 9000 seconds ($\Delta t = (9000 / 1.0) * 1$). The modelling employed in our manuscript uses a timestep of 3600s, which comfortably exceeds the CFL stability criterion. We now cite two additional papers which support this use of the CFL criterion to inform the choice of model time-stepping. L297-301.

Response to this point: The CFL condition is a reasonable choice for choosing a time-step, but it does not carry the same weight as the principle for which it was developed. The CFL condition was developed to ensure numerical stability of explicit schemes that solve partial differential equations for advection (e.g., $dv/dt + u dv/dx=0$) or (equivalently) a wave equation ($d^2z/dt^2 + cd^2z/dx^2=0$). If the CFL criterion appropriate for a particular explicit scheme is not met, the numerics blow up. Thus it is essential for numerical stability of these explicit schemes to ensure that this criterion is met. Otherwise, a solution is not possible.

In the case of Lagrangian simulations, the CFL criterion does not play a similar role. The trajectory will not blow up if the CFL criterion is not met. I acknowledge that it is a sensible first choice (one can imagine that excursion across multiple grid cells within a single time-step has obvious interpolation errors of the trajectory). But even advection within one grid cell in a time-step has discretization errors. Thus for Lagrangian calculations, there is a continuum of increasing accuracy as the time step decreases, not a sudden criterion that ensures it. The only way to test if a solution has converged is to carry out the calculation with 1/2 or twice the time-step and to see if the statistics of the trajectories are consistent. Each individual trajectory will be different with a different time-step.

The two references chosen (71,72) are biological analyses of Lagrangian floats that describe the CFL condition as though it ensures accuracy of the trajectories. This implication is now in this paper as well. I think this is misleading readers into

thinking that the CFL condition has been shown in numerical analysis of Lagrangian trajectories to ensure accurate results. I am not aware of such a numerical analysis.

I request that the authors adjust the discussion of stability. They can reference the CFL condition if they like, but should explicitly state that it has no formal bearing on solving ordinary differential equations like $dx/dt=u$, though it is reasonable.

I very much like the results in Supp. Fig 3, which satisfies me that the results are not sensitive to the time-step. Very nice.

Reviewer #3

(Remarks to the Author)

I was asked by the editor to assess authors' response to the comments from reviewer 1. I have interleaved my thoughts below and also provide additional comments at the end of the document.

Reviewer #1 (Remarks to the Author):

Review of "Host dispersal relaxes selective pressures in rafting microbiomes, triggering successional changes" by Pearman et al.

C1) This very well written manuscript describes changes in microbiome composition of seaweeds that have become detached and formed floating rafts in oceanic waters off Aotearoa, New Zealand, conducted a comprehensive comparison between rafting seaweeds and conspecifics remaining attached to the benthos in their 'native' / donor populations, and related differences to environmental variables and host genetics, using innovative analyses and the context of two community ecology hypotheses. They report an increase in disease-associated bacteria in rafting kelp and increased rates of dysbiosis in rafting microbiomes compared to non-raft kelp. These changes are correlated with time spent in raft (however this relationship appears non-linear with evidence of a 'return' to non-raft microbiome composition after longer periods spent drifting) and to a lesser extent, SST, with higher variability in SST correlating with dysbiosis and a loss of core microbiome taxa present in attached (non-raft) seaweed.

The data collected are very interesting and certainly contribute to our understanding of seaweed-associated microbiomes, providing new insights into how they might change when benthic seaweeds become detached but persist as hosts for long periods of time. The authors cite other articles that postulate raft algae such as the ones sampled in this study, could be vectors of disease and other microbes into new areas..

However, aspects of the sampling design are not clearly described in the manuscript and I am concerned that some of the conclusions are not well supported by the data and that other hypotheses or interpretations might be equally as valid. Specifically:

R1) We thank the reviewer for their comments, and are pleased that our manuscript was of interest to them. In response to the reviewer's concerns we now better detail our methods, and more clearly demonstrate that our conclusions are supported by the results presented. This clarification includes discussion of other hypotheses and interpretations and, where appropriate, our reasoned arguments in favour of some hypotheses over others. We detail responses to specific comments below.

C2) Time spent in raft environment: I appreciate the modelling approach used to estimate length of time individual seaweeds spent in the rafting environment and that it is difficult to obtain such data from sampling alone. However, this did appear to me to be a bit of a limitation in terms of interpreting the data presented here. Disease-associated bacteria are often also ecologically relevant as decomposers and often over-represented on older tissues, playing roles in natural senescence and microbial decomposition. A recurring question I asked while reviewing this manuscript was how you can know whether this is really disease or just natural processes? Without knowing how long the kelp have been rafting for, or their condition (e.g. visible signs of disease [these were mentioned once in the ms. but no data were presented], pigmentation, PAR, fouling, grazing, etc.), it's very difficult to separate disease and normal senescence related decomposition. Seaweed-associated microbiomes change dramatically with dramatic changes in environment and with host age and condition. Teasing these processes apart from rafting specifically, is difficult, but I appreciate the value in trying, in the context of dispersal.

R2) We agree with the reviewer that it is quite possible that many of our 'disease associated bacteria' are associated with decomposition or degradation more than disease per se. A more appropriate interpretation is that these taxa are broadly associated with poor condition (via disease or natural senescence/decay). As noted by R1 "Disease-associated bacteria are often also ecologically relevant as decomposers and often over-represented on older tissues, playing roles in natural senescence and microbial decomposition", and, when compiling our list of decomposition-associated bacteria (Table S2), taxa associated with these older/decaying tissues were identified (as opposed to taxa with a direct link to a specific disease). We now explicitly highlight that these taxa are decomposition-associated bacteria and now refer to them as such throughout the revised ms (L195-205).

New comments:

The response address the issue of disease versus decomposition function, however I have more addition concerns on the validity of the "DAB" classification as described below.

In the revised manuscript we have also clarified the drift modelling methods to better communicate that we do have an

estimate of rafting time for each collected raft (L325-326, L350). Furthermore we now also include reference to our recent work (Pearman 2024; Environmental Microbiology; L574-581) which examined the effects of age on microbiome composition and found only minimal age-related differences between samples.

New comments:

This is a satisfactory response.

C3) Similarly, it is difficult to assign causality to a positive relationship between SST (or variation therein) and dysbiosis in the microbiome (L458-460) as there are too many coincident, unexplored factors that could provide alternative explanations.

R3) We have modified the language of this point to clarify that, as noted by R1, SST can co-vary with a range of other unexplored factors (which builds on a similar point made by R2), and that a correlation between σ SST and dysbiosis does not imply causation. We frame our hypothesis more generally and now discuss alternative explanations for the patterns we described (L212-222, L615-618).

New comments:

This is a satisfactory response.

C4) Apparent assumption about return of raft algae to "target zone": At several points in the ms. (L269 – 283 and L532) it appears that there is an assumption or expectation that after some time spent in the drift, the kelp will return to a benthic lifestyle? Perhaps I am misinterpreting the authors' meaning, but kelp do not reattach after they become detached and, whilst sometimes remaining reproductively viable for some time, will either sink into deep ocean sediments, be consumed by herbivores or become part of the detrital food web via beach casting. I question whether this apparent assumption has influenced the authors interpretations of their data.

R4) We apologise for any confusion, we did not intend to suggest that that the algae return to the target zone. Our models are run backwards in time from the collection point of the raft (i.e., their last and only known position) from the time point of collection. Target zones were based on molecular data of both rafts and non rafts, as these data allowed to us to identify a putative mainland origin population for the rafts. We then ran models backwards in time looking at all possible trajectories that connected the collection point of the raft to the putative mainland origin. Therefore there is no assumption that rafts re-attach or return to a benthic life style, as noted by R1 rafts cannot re-attach.

In summary, the rafting timeline of a raft goes

Detachment from Source population -> Rafting -> Collection of raft offshore

However we conducted backwards in time modelling such that our models work as follows:

Collection of raft offshore -> Rafting (Reversed) -> Source population (target zone).

We have clarified this point on L321, L325-327, L332, L398, and L597-600.

New comments:

The manuscript is now clear on these points.

C5) The interpretation that having a healthy microbiome facilitates longer term dispersal is interesting (L535). Equally, though, microbiome 'healthiness' could simply be a function of seaweed condition, which is the result of multiple biological and physical factors.

R5) We agree that there are likely multiple biological and physical factors that could be affecting microbiome 'healthiness'. We now discuss these factors, and how our findings could be interpreted with this in mind (L603-608, L577-581). The concept that healthiness is a function of seaweed condition also aligns with our consideration of the bacteria in Table S2 as decomposition-associated bacteria rather than being specifically associated with disease. .

New comments:

I think the authors refer in their response to Table S3 rather than S2. I have concern about the classification of genera into the given functional category (see my additional comments below).

C6) Host genetics and microbiomes: Species-specific seaweed-associated microbiomes are typical, yet not observed here, where microbiomes were similar in two non-raft species of kelp showing no such patterns, according to 16S (L389-390; Fig. 3a). Yet later, when population-level genetics are assessed against microbiomes, there a strong relationship is reported (L439-442), with 21% and 12% of the variation in mircobiomes explained by host genetics on non-raft and raft kelp respectively. I don't know how to interpret this and this apparent discrepancy is not discussed further in the ms.

R6) We have removed this analysis from the manuscript, following re-analysis based on this comment we found that our raft genetic samples were dominated by host *D. antarctica*, while non-raft genetic samples were dominated by sister species *D. poha* thus making such a comparison difficult to justify from a genetic basis. Nevertheless, in our previous work we show that there are little to no differences in the microbiomes between *D. antarctica* and *D. poha*. Thus although we may normally expect to see distinct microbiomes between species of macroalgae, *D. poha* and *D. antarctica* are morphologically and genetically extremely similar, and occur in very similar environments. In our previous work (Pearman 2023 – Annals of Botany) we have shown that the relationships between host genetics and microbiome structure are typically relatively weak

for *Durvillaea* (both *D. poha* and *D. antarctica*), while environmental conditions are much more important. Thus, given the environmental overlap of both species, it is not surprising that we do not see evidence for species specific patterns. However, we modified Fig. 2c to indicate species of *Durvillaea* in question.

New comments:

This response seem to address the issue.

C7) Sampling design: How many replicate individuals were sampled per location along the transect? Was seawater also collected at non-raft locations? I could not see this reported in the Methods (but data appear in the Results). Which replicates were merged as per L112 and L152? Which species were sampled at which location? Although all the technical methodologies appear sound and appropriate to me, it is difficult to critically assess some of the analyses and conclusions without understanding the sampling design in more detail.

R7) A total of 37 separate rafts were collected. This reflects the total number of rafts encountered during ~45 hrs of survey time (7 surveys, 6-7 hours per survey). Because of the rarity of encountering rafts at sea, we sampled every raft we encountered. We have added these details to the revised manuscript (L111-114, L117-118). Seawater samples were collected both where we collected rafts and where we sampled non-raft populations (L123-24). As a result, replicate rafts were not collected per location. Instead replicate swabs per raft were collected, with two blade swabs, and a meristem sample. The two blade swabs were merged for the analysis presented in Fig 1, this is now discussed on lines L177-178. We have also added on L112 the average number of non-raft samples collected per population.

New comments:

The manuscript is now clear on these points.

Other minor comments / questions?:

C8) • L73 – How big is a raft of *Durvillaea* ? One individual? How large an area is 70 million rafts of *Durvillaea*?

R8) A raft of *Durvillaea* can vary from ~1m long as a single blade, or can consist of multiple individuals grouped together over a ~10-20m² area. We now include photos of kelp rafts as supplementary material. When sampling rafts, we only sampled genetic individuals (i.e., where coalescent holdfasts occur resulting in rafts of multiple genetic individuals 'stuck together', we only sampled a single individual) (L111-114).

Regarding the 70 million raft estimate this was originally estimated over the entire Southern Ocean based on rafts observed along a ship voyage from Hobart to Macquarie Island.

New comments:

This is a satisfactory response.

C9) • Intermediate Stochastic Hypothesis and Ana Karenina Hypothesis should be clearly defined in Introduction for non-expert readers outside of the authors' discipline. Alternatively, remove from the manuscript as they are inadequately explored or explained in the context of the data, as currently presented.

R9) We have added a paragraph to the introduction to expand and discuss these hypotheses in terms of understanding microbiomes. L79-97.

New comments:

This is a satisfactory response.

C10) • L94 - Diagram /further description indicating palmate meristem region of the thallus would benefit non-expert readers from outside of the authors' discipline.

R10) We have added additional discussion of what the palmate meristem refers to and added a reference for further information, the figure below shows the position of palmate meristem. L117-118.

New comments:

I could find this figure in the re-submitted version

C11) • When discussing how core microbiomes were identified and classified, it would be helpful to explain the proportion of samples in which microorganisms needed to be present to be considered 'core' (in addition to the statistical methods used).

R11) We have added this explanation, clarifying that both high abundance and occupancy are required – with occupancy of >80% being required for a core microbe, L190-193. We have also added a supplementary figure to highlight the structure of the core microbial community.L471-473.

New comments:

This has been clearly explained now.

C12) • Description of how DABs were identified and analysed is a bit confusing (L166 – L185. Perhaps the paragraphs are around the wrong way?

R12) We have added discussion of the identification of DABs, and swapped these paragraphs around. We also now include in the methods that we used the iCAMP process in analysis of the DABs, and that we tested for differences in abundance of DABs between rafts and non-rafts. L195-205, L220-221.

New comments:

The iCAMP is clear now.

C13) • L284 – is “Environmental Data” SST only? If so, consider just calling it SST?

R13) We have replaced the term here with SST following this suggestion. L341.

New comments:

This is a satisfactory response.

C14) • L289 – are “in-place kelps” the same as “non-raft kelps”? Please be consistent with labels.

R14) Yes, in-place kelps are the same as non-raft kelp, we have changed the term here to non-raft kelp. L346.

New comments:

This is a satisfactory response.

C15) • Explanation of drogue is not clear to me - if they have lost their drogue, are they driven by sea surface processes or are they driven by prevailing currents alone?

R15) Yes - if they have lost their drogue they are driven primarily by surface wind- and wave-driven currents. We have added extra explanation and citations to emphasize this point, L334-340.

New comments:

This is clear now.

C16) • L332-333 – what the “geographical range” is, is not clear from the text. Perhaps cite the relevant figure?

R16) Geographical range here refers to the range of populations from which rafts originated, e.g., since rafts were inferred to originate from the highlighted zones on Fig1a, we had only used non-raft microbiome samples from those zones for this analysis. However, following comment C6 above, we have removed this analysis and thus this paragraph has been removed from the manuscript.

New comments:

This is a satisfactory response.

C17) • L356-357 - So if the confidence that a particular raft came from a particular location was high, the likelihood of that raft returning to that location was lower?

R17) We apologize for the confusion, as the reviewer noted earlier, rafts are incredibly unlikely to return to a location. What we had attempted to convey in this sentence was simply that there was a broad range of inferred raft-times from the genetic origin to the point of collection. The reason that high confidence assignments tended to have fewer particles reaching them was because we could restrict the potential geographic source to a much narrower range (i.e. if we were confident that a raft was from a specific local population then the target zone was relatively small, but when we had low-confidence assignments the target zone encompassed a broader region covering multiple populations) – thus fewer particles would hit these locations. We have clarified this in the revised ms (L285-287).

New comments:

This is clear now.

Additional comments:

I have one major concern (in addition to a few minor ones listed below).

I am not convinced of classifying entire genera into DAB (see L195ff) as its well-known that species or strains of the same genus can have quite distinct interactions with seaweeds. For example, some *Phaeobacter* strains have indeed be shown to cause disease (reference 111), while others strains of this genus have been shown to prevent diseases (e.g. Li et al. ISME J 2022). Similarly, some *Pseudoalteromonas* strains (mislabeled in Table S3 as *Alteromonas*; see cited reference 120, Swabe et al.) have been isolated from degraded algal tissue, while other *Pseudoalteromonas* strains have been show to benefit macroalgal host through preventing biofouling (e.g. Bernbom et al. Applied and Environmental Microbiology 2011). Without species or strain-level resolution this type of classification is not valid. I would therefore strongly suggest to remove

this aspect from the manuscript.

I also note that the proportion of “DAB” found is very small, being less than 0.5% of the total community and only changing by roughly 0.1% between raft and non-raft samples (see Figure 6A). This would suggest that those changes are unlikely to make a major contribution to the overall functionality of the seaweed microbiome. To provide more solid evidence for changes in functionality (e.g. more pathogenic potential or degradation), other approaches (e.g. metagenomics etc.) would be required that would also provide information on the >99.5% of the community that was not classified as “DAB”.

I think the manuscript will still be interesting without the “DAB” aspect, but acknowledge that its removal will require substantial re-writing of the discussion (e.g. L553ff, L567ff, L584ff, L615ff)

L51 (and elsewhere): Please note that there is not such thing as a 16S amplicon as “S” stands for “Svedberg unit”. Please refer to the proper name of the gene (i.e. 16S rRNA gene) in reference to amplicon sequencing.

L75: It is unclear how references 20 and 22 support the specific statement here on rafting events.

L116: Was the artificial seawater sterile? If yes, then specify so.

L118: Figure 1 does not show a diagram of the thallus.

L154: Please note that this primer pair only amplifies the V4 region.

L157ff: Please provide the code for the bioinformatic analysis performed here.

L160: Define ASV.

L172: Please state if these additional 81 samples were also extracted and sequenced on the same way as the samples in this current study.

L185: Space after %.

L228: Define GuHCl as “Gu” refers to the element “göfofinium”, which I assume was not used here.

L224: Provide supplier or enzymatic units for the PstI restriction enzyme. The same applies subsequently for the T4 DNA ligase.

L389: Label the geographies mentioned here in Figure 1 as not everybody has a detailed knowledge on NZ’s regions.

Figure 1: It is unclear what region in map A the purple colour in panel C corresponds to.

L420: In this section it would be useful to state how many different ASVs were found and also to provide evidence that the sequencing effort was sufficient to capture most of the diversity in any given samples (e.g. by providing Good’s coverage values and/or rarefaction curves).

L518: “microbiome community” is a bit of a tautology as “microbiome” refers to a collection or community of microorganisms.

Version 2:

Reviewer comments:

Reviewer #2

(Remarks to the Author)

The authors have done a nice re-edit of their text describing time-step sensitivity, appropriately removing the discussion of CFL and including sensitivity tests. Their Supplemental Figure S4 is excellent and shows the appropriate insensitivity to time-step.

I have no further concerns and recommend the paper for publication.

Reviewer #3

(Remarks to the Author)

All my concerns have been addressed. I congratulate the authors on this really nice manuscript.

To aid in the review process, responses to reviewer comments are below with changes in the manuscript regarding each point commented appropriately alongside the tracked changes. Please note that line numbers below refer to line numbers in the 'simple mark-up' track changes document.

REVIEWER COMMENTS

Reviewer #1 (Remarks to the Author):

Review of "Host dispersal relaxes selective pressures in rafting microbiomes, triggering successional changes" by Pearman et al.

C1) This very well written manuscript describes changes in microbiome composition of seaweeds that have become detached and formed floating rafts in oceanic waters off Aotearoa, New Zealand, conducted a comprehensive comparison between rafting seaweeds and conspecifics remaining attached to the benthos in their 'native' / donor populations, and related differences to environmental variables and host genetics, using innovative analyses and the context of two community ecology hypotheses. They report an increase in disease-associated bacteria in rafting kelp and increased rates of dysbiosis in rafting microbiomes compared to non-raft kelp. These changes are correlated with time spent in raft (however this relationship appears non-linear with evidence of a 'return' to non-raft microbiome composition after longer periods spent drifting) and to a lesser extent, SST, with higher variability in SST correlating with dysbiosis and a loss of core microbiome taxa present in attached (non-raft) seaweed.

The data collected are very interesting and certainly contribute to our understanding of seaweed-associated microbiomes, providing new insights into how they might change when benthic seaweeds become detached but persist as hosts for long periods of time). The authors cite other articles that postulate raft algae such as the ones sampled in this study, could be vectors of disease and other microbes into new areas..

However, aspects of the sampling design are not clearly described in the manuscript and I am concerned that some of the conclusions are not well supported by the data and that other hypotheses or interpretations might be equally as valid. Specifically:

***R1)** We thank the reviewer for their comments, and are pleased that our manuscript was of interest to them. In response to the reviewer's concerns we now better detail our methods, and more clearly demonstrate that our conclusions are supported by the results presented. This clarification includes discussion of other hypotheses and interpretations and, where appropriate, our reasoned arguments in favour of some hypotheses over others. We detail responses to specific comments below.*

C2) Time spent in raft environment: I appreciate the modelling approach used to estimate length of time individual seaweeds spent in the rafting environment and that it is difficult to obtain such data from sampling alone. However, this did appear to me to be a bit of a limitation in terms of interpreting the data presented here. Disease-associated bacteria are often also ecologically relevant as decomposers and often over-represented on older tissues,

playing roles in natural senescence and microbial decomposition. A recurring question I asked while reviewing this manuscript was how you can know whether this is really disease or just natural processes? Without knowing how long the kelp have been rafting for, or their condition (e.g. visible signs of disease [these were mentioned once in the ms. but no data were presented], pigmentation, PAR, fouling, grazing, etc.), it's very difficult to separate disease and normal senescence related decomposition. Seaweed-associated microbiomes change dramatically with dramatic changes in environment and with host age and condition. Teasing these processes apart from rafting specifically, is difficult, but I appreciate the value in trying, in the context of dispersal.

R2) We agree with the reviewer that is quite possible that many of our 'disease associated bacteria' are associated with decomposition or degradation more than disease per se. A more appropriate interpretation is that these taxa are broadly associated with poor condition (via disease or natural senescence/decay). As noted by R1 "Disease-associated bacteria are often also ecologically relevant as decomposers and often over-represented on older tissues, playing roles in natural senescence and microbial decomposition", and, when compiling our list of decomposition-associated bacteria (Table S2), taxa associated with these older/decaying tissues were identified (as opposed to taxa with a direct link to a specific disease). We now explicitly highlight that these taxa are decomposition-associated bacteria and now refer to them as such throughout the revised ms (L195-205).

*In the revised manuscript we have also clarified the drift modelling methods to better communicate that we **do** have an estimate of rafting time for each collected raft (L325-326, L350). Furthermore we now also include reference to our recent work (Pearman 2024; Environmental Microbiology; L574-581) which examined the effects of age on microbiome composition and found only minimal age-related differences between samples.*

C3) Similarly, it is difficult to assign causality to a positive relationship between SST (or variation therein) and dysbiosis in the microbiome (L458-460) as there are too many coincident, unexplored factors that could provide alternative explanations.

R3) We have modified the language of this point to clarify that, as noted by R1, SST can covary with a range of other unexplored factors (which builds on a similar point made by R2), and that a correlation between sigmaSST and dysbiosis does not imply causation. We frame our hypothesis more generally and now discuss alternative explanations for the patterns we described (L212-222, L615-618).

C4) Apparent assumption about return of raft algae to "target zone": At several points in the ms. (L269 – 283 and L532) it appears that there is an assumption or expectation that after some time spent in the drift, the kelp will return to a benthic lifestyle? Perhaps I am misinterpreting the authors' meaning, but kelp do not reattach after they become detached and, whilst sometimes remaining reproductively viable for some time, will either sink into deep ocean sediments, be consumed by herbivores or become part of the detrital food web via beach casting. I question whether this apparent assumption has influenced the authors interpretations of their data.

R4) We apologise for any confusion, we did not intend to suggest that that the algae return to the target zone. Our models are run backwards in time from the collection point of the raft (i.e., their last and only known position) from the time point of collection. Target zones were based on molecular data of both rafts and non rafts, as these data allowed to us to identify a

putative mainland origin population for the rafts. We then ran models backwards in time looking at all possible trajectories that connected the collection point of the raft to the putative mainland origin. Therefore there is no assumption that rafts re-attach or return to a benthic life style, as noted by R1 rafts cannot re-attach.

In summary, the rafting timeline of a raft goes

Detachment from Source population -> Rafting -> Collection of raft offshore

However we conducted backwards in time modelling such that our models work as follows:

Collection of raft offshore -> Rafting (Reversed) -> Source population (target zone).

We have clarified this point on L321, L325-327, L332, L398, and L597-600.

C5) The interpretation that having a healthy microbiome facilitates longer term dispersal is interesting (L535). Equally, though, microbiome ‘healthiness’ could simply be a function of seaweed condition, which is the result of multiple biological and physical factors.

R5) We agree that there are likely multiple biological and physical factors that could be affecting microbiome ‘healthiness’. We now discuss these factors, and how our findings could be interpreted with this in mind (L603-608, L577-581). The concept that healthiness is a function of seaweed condition also aligns with our consideration of the bacteria in Table S2 as decomposition-associated bacteria rather than being specifically associated with disease. .

C6) Host genetics and microbiomes: Species-specific seaweed-associated microbiomes are typical, yet not observed here, where microbiomes were similar in two non-raft species of kelp showing no such patterns, according to 16S (L389-390; Fig. 3a). Yet later, when population-level genetics are assessed against microbiomes, there a strong relationship is reported (L439-442), with 21% and 12% of the variation in microbiomes explained by host genetics on non-raft and raft kelp respectively. I don’t know how to interpret this and this apparent discrepancy is not discussed further in the ms.

*R6) We have removed this analysis from the manuscript, following re-analysis based on this comment we found that our raft genetic samples were dominated by host *D. antarctica*, while non-raft genetic samples were dominated by sister species *D. poha* thus making such a comparison difficult to justify from a genetic basis. Nevertheless, in our previous work we show that there are little to no differences in the microbiomes between *D. antarctica* and *D. poha*. Thus although we may normally expect to see distinct microbiomes between species of macroalgae, *D. poha* and *D. antarctica* are morphologically and genetically extremely similar, and occur in very similar environments. In our previous work (Pearman 2023 – *Annals of Botany*) we have shown that the relationships between host genetics and microbiome structure are typically relatively weak for *Durvillaea* (both *D. poha* and *D. antarctica*), while environmental conditions are much more important. Thus, given the environmental overlap of both species, it is not surprising that we do not see evidence for species specific patterns. However, we modified Fig. 2c to indicate species of *Durvillaea* in question.*

C7) Sampling design: How many replicate individuals were sampled per location along the transect? Was seawater also collected at non-raft locations? I could not see this reported in the Methods (but data appear in the Results). Which replicates were merged as per L112 and L152? Which species were sampled at which location? Although all the technical methodologies appear sound and appropriate to me, it is difficult to critically assess some of

the analyses and conclusions without understanding the sampling design in more detail.

R7) A total of 37 separate rafts were collected. This reflects the total number of rafts encountered during ~45 hrs of survey time (7 surveys, 6-7 hours per survey). Because of the rarity of encountering rafts at sea, we sampled every raft we encountered. We have added these details to the revised manuscript (L111-114, L117-118). Seawater samples were collected both where we collected rafts and where we sampled non-raft populations (L123-24). As a result, replicate rafts were not collected per location. Instead replicate swabs per raft were collected, with two blade swabs, and a meristem sample. The two blade swabs were merged for the analysis presented in Fig 1, this is now discussed on lines L177-178. We have also added on L112 the average number of non-raft samples collected per population.

Other minor comments / questions?:

C8) • L73 – How big is a raft of *Durvillaea* ? One individual? How large an area is 70 million rafts of *Durvillaea*?

*R8) A raft of *Durvillaea* can vary from ~1m long as a single blade, or can consist of multiple individuals grouped together over a ~10-20m² area. We now include photos of kelp rafts as supplementary material. When sampling rafts, we only sampled genetic individuals (i.e., where coalescent holdfasts occur resulting in rafts of multiple genetic individuals ‘stuck together’, we only sampled a single individual) (L111-114).*

Regarding the 70 million raft estimate this was originally estimated over the entire Southern Ocean based on rafts observed along a ship voyage from Hobart to Macquarie Island.

C9) • Intermediate Stochastic Hypothesis and Ana Karenina Hypothesis should be clearly defined in Introduction for non-expert readers outside of the authors’ discipline. Alternatively, remove from the manuscript as they are inadequately explored or explained in the context of the data, as currently presented.

R9) We have added a paragraph to the introduction to expand and discuss these hypotheses in terms of understanding microbiomes. L79-97.

C10) • L94 - Diagram /further description indicating palmate meristem region of the thallus would benefit non-expert readers from outside of the authors’ discipline.

R10) We have added additional discussion of what the palmate meristem refers to and added a reference for further information, the figure below shows the position of palmate meristem. L117-118.

C11) • When discussing how core microbiomes were identified and classified, it would be helpful to explain the proportion of samples in which microorganisms needed to be present to be considered ‘core’ (in addition to the statistical methods used).

R11) We have added this explanation, clarifying that both high abundance and occupancy are required – with occupancy of >80% being required for a core microbe, L190-193. We have also added a supplementary figure to highlight the structure of the core microbial community. L471-473.

C12) • Description of how DABs were identified and analysed is a bit confusing (L166 – L185. Perhaps the paragraphs are around the wrong way?

R12) We have added discussion of the identification of DABs, and swapped these paragraphs around. We also now include in the methods that we used the iCAMP process in analysis of the DABs, and that we tested for differences in abundance of DABs between rafts and non-rafts. L195-205, L220-221.

C13) • L284 – is “Environmental Data” SST only? If so, consider just calling it SST?

R13) We have replaced the term here with SST following this suggestion. L341.

C14) • L289 – are “in-place kelps” the same as “non-raft kelps”? Please be consistent with labels.

R14) Yes, in-place kelps are the same as non-raft kelp, we have changed the term here to non-raft kelp. L346.

C15) • Explanation of drogue is not clear to me - if they have lost their drogue, are they driven by sea surface processes or are they driven by prevailing currents alone?

R15) Yes - if they have lost their drogue they are driven primarily by surface wind- and wave-driven currents. We have added extra explanation and citations to emphasize this point, L334-340.

C16) • L332-333 – what the “geographical range” is, is not clear from the text. Perhaps cite the relevant figure?

R16) Geographical range here refers to the range of populations from which rafts originated, e.g., since rafts were inferred to originate from the highlighted zones on Fig1a, we had only used non-raft microbiome samples from those zones for this analysis. However, following comment C6 above, we have removed this analysis and thus this paragraph has been removed from the manuscript.

C17) • L356-357 - So if the confidence that a particular raft came from a particular location was high, the likelihood of that raft returning to that location was lower?

R17) We apologize for the confusion, as the reviewer noted earlier, rafts are incredibly unlikely to return to a location. What we had attempted to convey in this sentence was simply that there was a broad range of inferred raft-times from the genetic origin to the point of collection. The reason that high confidence assignments tended to have fewer particles

reaching them was because we could restrict the potential geographic source to a much narrower range (i.e. if we were confident that a raft was from a specific local population then the target zone was relatively small, but when we had low-confidence assignments the target zone encompassed a broader region covering multiple populations) – thus fewer particles would hit these locations. We have clarified this in the revised ms (L285-287).

Reviewer #2 (Remarks to the Author):

C18) Below is my review of "Host dispersal relaxes selective pressures in rafting microbiomes, triggering successional changes" by Pearman et al. I was asked to focus on the particle modeling specifically, and that is the main emphasis of my review.

The manuscript describes analysis of microbiomes taken from kelp blades from two populations in waters near southern New Zealand in 2021 and 2022. One group had drifted with ocean conditions (currents, waves, and winds) and is referred to as rafted. The other samples, non-rafts, came from established populations in nearshore environments, and thus did not experience variable open ocean conditions. Ensembles of trajectories of rafts were estimated using a particle tracking model, driven by currents from ocean reanalyses and winds from an atmospheric product. Sea surface temperature (SST) was sampled along trajectories from a data product to obtain SST variability possibly experienced along the track. Differences in microbiomes and microbiome diversity were established between the populations. A model suggests that dissimilarity increased with raft duration. A model of microbiome diversity richness showed greatest richness for intermediate values of SST variability, less dependent on duration.

The paper is well-written and results are interesting to ecological and oceanographic communities. Conclusions are well founded based on results obtained. The particle tracking modeling is state-of-the-art, with appropriate, high quality input fields for currents, winds, and SST. I recommend publication with minor modifications.

R18) We thank the reviewer for their comments.

Larger comments

C19) 1) The authors should include information relating to sensitivity tests done with the trajectory time-stepping. For example, had the authors chosen time-steps one half or one quarter as large as those used, would the statistics of their results change? I do not believe any information about time-stepping is presently included in the manuscript but should be. I assume that these sensitivity tests were carried out and reasonable convergence in statistics was achieved. The authors could confirm this.

R19) We now include in the supplementary data a range of sensitivity tests to both the trajectory time-stepping and the horizontal diffusivity. Although the sensitivity tests show that our results are not affected by a change in time step, we also highlight that our chosen timestep meets the Courant-Friedrichs-Lewy (CFL) stability criterion. The CFL criterion is:

$$\Delta t = (dx / V_{max}) * C$$

where dx is the model grid resolution in m, V_{max} is the maximum velocity anticipated in a grid cell, C is a constant = 1 to meet the CFL criteria and Δt is the minimum time step necessary to meet the CFL stability criterion.

The GLORYS12 ocean model used in our manuscript has a spatial resolution(dx) of $1/12^\circ$ lat-lon, which is approximately 9 km at the latitude of New Zealand. In the vicinity of southern New Zealand / Tasman Sea, velocities in GLORYS grid cells are typically < 0.5 m/s. They get slightly larger in the core of the Southland Current, inshore of which some of our sampling took place. but even then, they are < 1.0 m/s (see e.g. <https://data.marine.copernicus.eu/-/hj20k5w1bt>).

Using a conservative value of 1.0 m/s for V_{max} provides a minimum Δt required to meet the CFL criterion of 9000 seconds ($\Delta t = (9000 / 1.0) * 1$). The modelling employed in our manuscript uses a timestep of 3600s, which comfortably exceeds the CFL stability criterion. We now cite two additional papers which support this use of the CFL criterion to inform the choice of model time-stepping. L297-301.

C20) 2) SST variability is treated as a likely cause for stochastic forcing. The authors might speculate on whether temperature variability is likely the driving factor or if it is a proxy for mixing with different water masses with different biological or nutrient environments. I would have thought the greater influence would be mixing with different water masses with exposure to different communities than temperature variability alone. The authors might mention for example whether the SST variability is a proxy for other factors or if, in their opinion, it is the temperature variability that is critical.

R20) Thank you, we have now increased discussion around this point – emphasizing that SST variability is perhaps better treated as a proxy for other factors as well. This is particularly important because many other environmental factors covary with SST, especially those for which we do not have high resolution and robust spatio-temporal data. L528-546

Minor comments

C21) 3) Line 11: don't -> do not
R21) Changed.

C22) 4) Line 257: "randomly seeded laterally within a 1 km radius of the collection coordinates."

R22) Corrected.

C23) 5) Line 264: I request a bit more information about PIW-1, PIW-5, and PIW-6. These are available in the opendrift documentation, but I think information here can be included, specifically what PIW stands for and what the numbers indicate.

R23)

We have added the requested information to the revised ms (L309-317) and now include a table (Table S2) with the numeric values used to parameterise these particles.

C24) 6) Line 357: "reached the source location inferred via genetic analysis through

backward in time advection, with a median...

R24) We have modified this sentence as requested.

C25) 7) Line 359: replace target with source (source is clearer in this context).

R25) We have made this change.

REVIEWER COMMENTS

Reviewer #2 (Remarks to the Author):

Below is my review of the revision to "Host dispersal relaxes selective pressures in rifting microbiomes, triggering successional changes" by Pearman et al.

C1) The authors have done a good job of addressing my points from the first review. However, I disagree with one point and would like the authors to correct what I view is now a growing misperception.

C19) 1) The authors should include information relating to sensitivity tests done with the trajectory time-stepping. For example, had the authors chosen time-steps one half or one quarter as large as those used, would the statistics of their results change? I do not believe any information about time-stepping is presently included in the manuscript but should be. I assume that these sensitivity tests were carried out and reasonable convergence in statistics was achieved. The authors could confirm this.

Round1_R19) We now include in the supplementary data a range of sensitivity tests to both the trajectory time-stepping and the horizontal diffusivity. Although the sensitivity tests show that our results are not affected by a change in time step, we also highlight that our chosen timestep meets

the Courant-Friedrichs-Lewy (CFL) stability criterion. The CFL criterion is:

$$\Delta t = (dx / V_{max}) * C$$

where dx is the model grid resolution in m, V max is the maximum velocity anticipated in a grid cell, C is a constant = 1 to meet the CFL criteria and Δt is the minimum time step necessary to meet the CFL stability criterion.

The GLORYS12 ocean model used in our manuscript has a spatial resolution(dx) of 1/12° lat-lon, which is approximately 9 km at the latitude of New Zealand. In the vicinity of southern New Zealand / Tasman Sea, velocities in GLORYS grid cells are typically < 0.5 m/s. They get slightly larger in the core of the Southland Current, inshore of which some of our sampling took place. but even then, they are < 1.0 m/s (see e.g. <https://data.marine.copernicus.eu/-/hj20k5w1bt>).

*Using a conservative value of 1.0 m/s for Vmax provides a minimum Δt required to meet the CFL criterion of 9000 seconds (Δt = (9000 / 1.0) * 1). The modelling employed in our manuscript uses a timestep of 3600s, which comfortably exceeds the CFL stability criterion. We now cite two additional papers which support this use of the CFL criterion to inform the choice of model time-stepping. L297-301.*

Response to this point: The CFL condition is a reasonable choice for choosing a time-step, but it does not carry the same weight as the principle for which it was developed. The CFL condition was developed to ensure numerical stability of explicit schemes that solve partial differential equations for advection (e.g., $dv/dt + udv/dx=0$) or (equivalently) a wave equation ($d^2z/dt^2 + cd^2z/dx^2=0$). If the CFL criterion appropriate for a particular explicit scheme is not met, the numerics blow up. Thus it is essential for numerical stability of these explicit schemes to ensure that this criterion is met. Otherwise, a solution is not possible.

In the case of Lagrangian simulations, the CFL criterion does not play a similar role. The trajectory will not blow up if the CFL criterion is not met. I acknowledge that it is a sensible first choice (one can imagine that excursion across multiple grid cells within a single time-step has obvious interpolation errors of the trajectory). But even advection within one grid cell in a a time-step has discretization errors. Thus for Lagrangian calculations, there is a continuum of increasing accuracy as the time step decreases, not a sudden criterion that ensures it. The only way to test if a solution has converged is to carry out the calculation with 1/2 or twice the time-step and to see if the statistics of the trajectories are consistent. Each individual trajectory will be different with a different time-step.

The two references chosen (71,72) are biological analyses of Lagrangian floats that describe the CFL condition as though it ensures accuracy of the trajectories. This implication is now in this paper as well. I think this is misleading readers into thinking that the CFL condition has been shown in numerical analysis of Lagrangian trajectories to ensure accurate results. I am not aware of such a numerical analysis.

I request that the authors adjust the discussion of stability. They can reference the CFL condition if they like, but should explicitly state that it has no formal bearing on solving ordinary differential equations like $dx/dt=u$, though it is reasonable.

I very much like the results in Supp. Fig 3, which satisfies me that the results are not sensitive to the time-step. Very nice.

R1) We appreciate the reviewers comments here. We have adjusted the discussion of stability to remove explicit reference to the CFL condition and reflect the emphasis on the sensitivity tests performed with different trajectory time-stepping. See lines 285-291.

Reviewer #3 (Remarks to the Author):

C2) I was asked by the editor to assess authors' response to the comments from reviewer 1. I have interleaved my thoughts below and also provide additional comments at the end of the document.

R2) We thank the reviewer for their helpful comments.

Reviewer #1 (Remarks to the Author):

Review of "Host dispersal relaxes selective pressures in rafting microbiomes, triggering successional changes" by Pearman et al.

C1) This very well written manuscript describes changes in microbiome composition of seaweeds that have become detached and formed floating rafts in oceanic waters off Aotearoa, New Zealand, conducted a comprehensive comparison between rafting seaweeds and conspecifics remaining attached to the benthos in their 'native' / donor populations, and related differences to environmental variables and host genetics, using innovative analyses and the context of two community ecology hypotheses. They report an increase in disease-associated bacteria in rafting kelp and increased rates of dysbiosis in rafting microbiomes compared to non-raft kelp. These changes are correlated with time spent in raft (however this

relationship appears non-linear with evidence of a 'return' to non-raft microbiome composition after longer periods spent drifting) and to a lesser extent, SST, with higher variability in SST correlating with dysbiosis and a loss of core microbiome taxa present in attached (non-raft) seaweed.

The data collected are very interesting and certainly contribute to our understanding of seaweed-associated microbiomes, providing new insights into how they might change when benthic seaweeds become detached but persist as hosts for long periods of time). The authors cite other articles that postulate raft algae such as the ones sampled in this study, could be vectors of disease and other microbes into new areas..

However, aspects of the sampling design are not clearly described in the manuscript and I am concerned that some of the conclusions are not well supported by the data and that other hypotheses or interpretations might be equally as valid. Specifically:

R1) We thank the reviewer for their comments, and are pleased that our manuscript was of interest to them. In response to the reviewer's concerns we now better detail our methods, and more clearly demonstrate that our conclusions are supported by the results presented. This clarification includes discussion of other hypotheses and interpretations and, where appropriate, our reasoned arguments in favour of some hypotheses over others. We detail responses to specific comments below.

C2) Time spent in raft environment: I appreciate the modelling approach used to estimate length of time individual seaweeds spent in the rafting environment and that it is difficult to obtain such data from sampling alone. However, this did appear to me to be a bit of a limitation in terms of interpreting the data presented here. Disease-associated bacteria are often also ecologically relevant as decomposers and often over-represented on older tissues, playing roles in natural senescence and microbial decomposition. A recurring question I asked while reviewing this manuscript was how you can know whether this is really disease or just natural processes? Without knowing how long the kelp have been rafting for, or their condition (e.g. visible signs of disease [these were mentioned once in the ms. but no data were presented], pigmentation, PAR, fouling, grazing, etc.), it's very difficult to separate disease and normal senescence related decomposition. Seaweed-associated microbiomes change dramatically with dramatic changes in environment and with host age and condition. Teasing these processes apart from rafting specifically, is difficult, but I appreciate the value in trying, in the context of dispersal.

R2) We agree with the reviewer that is quite possible that many of our 'disease associated bacteria' are associated with decomposition or degradation more than disease per se. A more appropriate interpretation is that these taxa are broadly associated with poor condition (via disease or natural senescence/decay). As noted by R1 "Disease-associated bacteria are often also ecologically relevant as decomposers and often over-represented on older tissues, playing roles in natural senescence and microbial decomposition", and, when compiling our list of decomposition-associated bacteria (Table S2), taxa associated with these older/decaying tissues were identified (as opposed to taxa with a direct link to a specific disease). We now explicitly highlight that these taxa are decomposition-associated bacteria and now refer to them as such throughout the revised ms (L195-205).

New comments:

C3) The response address the issue of disease versus decomposition function, however I have more addition concerns on the validity of the "DAB" classification as decribed below.

R3) We address this comment in response to C6

In the revised manuscript we have also clarified the drift modelling methods to better communicate that we do have an estimate of rafting time for each collected raft (L325-326, L350). Furthermore we now also include reference to our recent work (Pearman 2024; Environmental Microbiology; L574-581) which examined the effects of age on microbiome composition and found only minimal age-related differences between samples.

New comments:

This is a satisfactory response.

C3) Similarly, it is difficult to assign causality to a positive relationship between SST (or variation therein) and dysbiosis in the microbiome (L458-460) as there are too many coincident, unexplored factors that could provide alternative explanations.

R3) We have modified the language of this point to clarify that, as noted by R1, SST can co-vary with a range of other unexplored factors (which builds on a similar point made by R2), and that a correlation between sigmaSST and dysbiosis does not imply causation. We frame our hypothesis more generally and now discuss alternative explanations for the patterns we described (L212-222, L615-618).

New comments:

This is a satisfactory response.

C4) Apparent assumption about return of raft algae to “target zone”: At several points in the ms. (L269 – 283 and L532) it appears that there is an assumption or expectation that after some time spent in the drift, the kelp will return to a benthic lifestyle? Perhaps I am misinterpreting the authors’ meaning, but kelp do not reattach after they become detached and, whilst sometimes remaining reproductively viable for some time, will either sink into deep ocean sediments, be consumed by herbivores or become part of the detrital food web via beach casting. I question whether this apparent assumption has influenced the authors interpretations of their data.

R4) We apologise for any confusion, we did not intend to suggest that that the algae return to the target zone. Our models are run backwards in time from the collection point of the raft (i.e., their last and only known position) from the time point of collection. Target zones were based on molecular data of both rafts and non rafts, as these data allowed to us to identify a putative mainland origin population for the rafts. We then ran models backwards in time looking at all possible trajectories that connected the collection point of the raft to the putative mainland origin. Therefore there is no assumption that rafts re-attach or return to a benthic life style, as noted by R1 rafts cannot re-attach.

In summary, the rafting timeline of a raft goes

*Detachment from Source population -> Rafting -> Collection of raft offshore
However we conducted backwards in time modelling such that our models work as follows:*

Collection of raft offshore -> Rafting (Reversed) -> Source population (target zone).

We have clarified this point on L321, L325-327, L332, L398, and L597-600.

New comments:

The manuscript is now clear on these points.

C5) The interpretation that having a healthy microbiome facilitates longer term dispersal is interesting (L535). Equally, though, microbiome ‘healthiness’ could simply be a function of seaweed condition, which is the result of multiple biological and physical factors.

R5) We agree that there are likely multiple biological and physical factors that could be affecting microbiome ‘healthiness’. We now discuss these factors, and how our findings could be interpreted with this in mind (L603-608, L577-581). The concept that healthiness is a function of seaweed condition also

aligns with our consideration of the bacteria in Table S2 as decomposition-associated bacteria rather than being specifically associated with disease. .

New comments:

C4) I think the authors refer in their response to Table S3 rather than S2. I have concern about the classification of genera into the given functional category (see my additional comments below).

R4) Please see response to C6.

C6) Host genetics and microbiomes: Species-specific seaweed-associated microbiomes are typical, yet not observed here, where microbiomes were similar in two non-raft species of kelp showing no such patterns, according to 16S (L389-390; Fig. 3a). Yet later, when population-level genetics are assessed against microbiomes, there a strong relationship is reported (L439-442), with 21% and 12% of the variation in mircobiomes explained by host genetics on non-raft and raft kelp respectively. I don't know how to interpret this and this apparent discrepancy is not discussed further in the ms.

*R6) We have removed this analysis from the manuscript, following re-analysis based on this comment we found that our raft genetic samples were dominated by host *D. antarctica*, while non-raft genetic samples were dominated by sister species *D. poha* thus making such a comparison difficult to justify from a genetic basis. Nevertheless, in our previous work we show that there are little to no differences in the microbiomes between *D. antarctica* and *D. poha*. Thus although we may normally expect to see distinct microbiomes between species of macroalgae, *D. poha* and *D. antarctica* are morphologically and genetically extremely similar, and occur in very similar environments. In our previous work (Pearman 2023 – *Annals of Botany*) we have shown that the relationships between host genetics and microbiome structure are typically relatively weak for *Durvillaea* (both *D. poha* and *D. antarctica*), while environmental conditions are much more important. Thus, given the environmental overlap of both species, it is not surprising that we do not see evidence for species specific patterns. However, we modified Fig. 2c to indicate species of *Durvillaea* in question.*

New comments:

This response seem to address the issue.

C7) Sampling design: How many replicate individuals were sampled per location along the transect? Was seawater also collected at non-raft

locations? I could not see this reported in the Methods (but data appear in the Results). Which replicates were merged as per L112 and L152? Which species were sampled at which location? Although all the technical methodologies appear sound and appropriate to me, it is difficult to critically assess some of the analyses and conclusions without understanding the sampling design in more detail.

R7) A total of 37 separate rafts were collected. This reflects the total number of rafts encountered during ~45 hrs of survey time (7 surveys, 6-7 hours per survey). Because of the rarity of encountering rafts at sea, we sampled every raft we encountered. We have added these details to the revised manuscript (L111-114, L117-118). Seawater samples were collected both where we collected rafts and where we sampled non-raft populations (L123-24). As a result, replicate rafts were not collected per location. Instead replicate swabs per raft were collected, with two blade swabs, and a meristem sample. The two blade swabs were merged for the analysis presented in Fig 1, this is now discussed on lines L177-178. We have also added on L112 the average number of non-raft samples collected per population.

New comments:

The manuscript is now clear on these points.

Other minor comments / questions?:

C8) • L73 – How big is a raft of Durvillaea ? One individual? How large an area is 70 million rafts of Durvillaea?

R8) A raft of Durvillaea can vary from ~1m long as a single blade, or can consist of multiple individuals grouped together over a ~10-20m² area. We now include photos of kelp rafts as supplementary material. When sampling rafts, we only sampled genetic individuals (i.e., where coalescent holdfasts occur resulting in rafts of multiple genetic individuals ‘stuck together’, we only sampled a single individual) (L111-114).

Regarding the 70 million raft estimate this was originally estimated over the entire Southern Ocean based on rafts observed along a ship voyage from Hobart to Macquarie Island.

New comments:

This is a satisfactory response.

C9) • Intermediate Stochastic Hypothesis and Ana Karenina Hypothesis

should be clearly defined in Introduction for non-expert readers outside of the authors' discipline. Alternatively, remove from the manuscript as they are inadequately explored or explained in the context of the data, as currently presented.

R9) We have added a paragraph to the introduction to expand and discuss these hypotheses in terms of understanding microbiomes. L79-97.

New comments:

This is a satisfactory response.

C10) • L94 - Diagram /further description indicating palmate meristem region of the thallus would benefit non-expert readers from outside of the authors' discipline.

R10) We have added additional discussion of what the palmate meristem refers to and added a reference for further information, the figure below shows the position of palmate meristem. L117-118.

New comments:

C5) I could find this figure in the re-submitted version

R5) We have added this to Supp. Fig. 1 to clarify.

C11) • When discussing how core microbiomes were identified and classified, it would be helpful to explain the proportion of samples in which microorganisms needed to be present to be considered 'core' (in addition to the statistical methods used).

R11) We have added this explanation, clarifying that both high abundance and occupancy are required – with occupancy of >80% being required for a core microbe, L190-193. We have also added a supplementary figure to highlight the structure of the core microbial community.L471-473.

New comments:

This has been clearly explained now.

C12) • Description of how DABs were identified and analysed is a bit confusing (L166 – L185. Perhaps the paragraphs are around the wrong way?

R12) We have added discussion of the identification of DABs, and swapped these paragraphs around. We also now include in the methods that we used the iCAMP process in analysis of the DABs, and that we tested for differences in abundance of DABs between rafts and non-rafts. L195-205, L220-221.

New comments:

The iCAMP is clear now.

C13) • L284 – is “Environmental Data” SST only? If so, consider just calling it SST?

R13) We have replaced the term here with SST following this suggestion. L341.

New comments:

This is a satisfactory response.

C14) • L289 – are “in-place kelps” the same as “non-raft kelps”? Please be consistent with labels.

R14) Yes, in-place kelps are the same as non-raft kelp, we have changed the term here to non-raft kelp. L346.

New comments:

This is a satisfactory response.

C15) • Explanation of drogue is not clear to me - if they have lost their drogue, are they driven by sea surface processes or are they driven by prevailing currents alone?

R15) Yes - if they have lost their drogue they are driven primarily by surface wind- and wave-driven currents. We have added extra explanation and citations to emphasize this point, L334-340.

New comments:

This is clear now.

C16) • L332-333 – what the “geographical range” is, is not clear from the text. Perhaps cite the relevant figure?

R16) Geographical range here refers to the range of populations from which rafts originated, e.g., since rafts were inferred to originate from the highlighted zones on Fig1a, we had only used non-raft microbiome samples from those

zones for this analysis. However, following comment C6 above, we have removed this analysis and thus this paragraph has been removed from the manuscript.

New comments:

This is a satisfactory response.

C17) • L356-357 - So if the confidence that a particular raft came from a particular location was high, the likelihood of that raft returning to that location was lower?

R17) We apologize for the confusion, as the reviewer noted earlier, rafts are incredibly unlikely to return to a location. What we had attempted to convey in this sentence was simply that there was a broad range of inferred raft-times from the genetic origin to the point of collection. The reason that high confidence assignments tended to have fewer particles reaching them was because we could restrict the potential geographic source to a much narrower range (i.e. if we were confident that a raft was from a specific local population then the target zone was relatively small, but when we had low-confidence assignments the target zone encompassed a broader region covering multiple populations) – thus fewer particles would hit these locations. We have clarified this in the revised ms (L285-287).

New comments:

This is clear now.

Additional comments:

C6) I have one major concern (in addition to a few minor ones listed below).

I am not convinced of classifying entire genera into DAB (see L195ff) as its well-known that species or strains of the same genus can have quite distinct interactions with seaweeds. For example, some *Phaeobacter* strains have indeed be shown to cause disease (reference 111), while others strains of this genus have been shown to prevent diseases (e.g. Li et al. ISME J 2022). Similarly, some *Pseudoalteromonas* strains (mislabelled in Table S3 as *Alteromonas*; see cited reference 120, Swabe et al.) have been isolated from degraded algal tissue, while other *Pseudoalteromonas* strains have been show to benefit macroagal host through preventing biofouling (e.g. Bernbom et al. Applied and Environmental Microbiology 2011). Without species or strain-level resolution this type of classification is not valid. I would therefore strongly suggest to remove this aspect from the manuscript.

I also note that the proportion of “DAB” found is very small, being less than 0.5% of the total community and only changing by roughly 0.1% between raft and non-raft

samples (see Figure 6A). This would suggest that those changes are unlikely to make a major contribution to the overall functionality of the seaweed microbiome. To provide more solid evidence for changes in functionality (e.g. more pathogenic potential or degradation), other approaches (e.g. metagenomics etc.) would be required that would also provide information on the >99.5% of the community that was not classified as “DAB”.

I think the manuscript will still be interesting without the “DAB” aspect, but acknowledge that its removal will require substantial re-writing of the discussion (e.g. L553ff, L567ff, L584ff, L615ff)

R6) *We appreciate the reviewer’s comments and detailed consideration of this section. Following the reviewer’s comments we have removed discussion of the DABs all together. For example, see lines 475-495.*

C7) L51 (and elsewhere): Please note that there is not such thing as a 16S amplicon as “S” stands for “Svedberg unit”. Please refer to the proper name of the gene (i.e. 16S rRNA gene) in reference to amplicon sequencing.

R7) *We have corrected this through the manuscript (e.g., L77 and L152).*

C8) L75: It is unclear how references 20 and 22 support the specific statement here on rafting events.

R8) *We apologise for the mistake here, these two references were inserted manually and not updated automatically – the correct references are 21 and 23, which refer to papers which have conducted these rafting genomic analyses. See L102.*

C9) L116: Was the artificial seawater sterile? If yes, then specify so.

R9) *The water was sterile, this is now stated on line 115.*

C10) L118: Figure 1 does not show a diagram of the thallus.

R10) *We had meant to point the reader to Figure 1 of another manuscript, though we realise this was not at all clear – we now include, in Supp Fig. 1, a picture of the thallus and the sections in question. See L117.*

C11) L154: Please note that this primer pairs only amplifies the V4 region.

R11) We have corrected this, L155.

C12) L157ff: Please provide the code for the bioinformatic analysis performed here.

R12) We now point to our github repository here – L166.

C13) L160: Define ASV.

R13) Now defined, L160.

C14) L172: Please state if these additional 81 samples were also extracted and sequenced on the same way as the samples in this current study.

R14) We now state that these samples were processed identically, L175-176.

C15) L185: Space after %.

R15) Corrected, L187.

C16) L228: Define GuHCl as “Gu” refers to the element “goofinium”, which I assume was not used here.

R16) We now clarify the GuHCl refers to Guanidine Hydrochloride. L215.

C17) L224: Provide supplier or enzymatic units for the PstI restriction enzyme. The same applies subsequently for the T4 DNA ligase.

R17) We now specify the supplier and catalogue numbers here, L231, 234-235.

C18) L389: Label the geographies mention here in Figure 1 as not everybody has a detailed knowledge on NZ's regions.

R18) We have adjusted Figure 1 to highlight the mentioned locations.

C19) Figure 1: It is unclear what region in map A the purple colour in panel C corresponds to.

R19) *We have replaced the sections in question with multicoloured regions to emphasize the coloured sections. L134-135.*

C20) L420: In this section it would be useful to state how many different ASV were found and also to provide evidence that the sequencing effort was sufficient to capture most of the diversity in any given samples (e.g. by providing Good's coverage values and/or rarefaction curves).

R20) *We now highlight the pre- and post-rarefaction ASV counts, and direct the reader to the rarefaction curves – L410-412. Following advice from the developer of DADA2 we do not report Good's coverage value in the manuscript, however for completeness in the review process we provide it here - <https://github.com/benjjneb/dada2/issues/1374>. We had a mean Goods coverage index of 0.99 with a standard deviation of 0.01.*

C21) L518: "microbiome community" is a bit of a tautology as "microbiome" refers to a collection or community of microorganisms.

R21) *We have removed the term community here – L499.*

REVIEWERS' COMMENTS

Reviewer #2 (Remarks to the Author):

C1) The authors have done a nice re-edit of their text describing time-step sensitivity, appropriately removing the discussion of CFL and including sensitivity tests. Their Supplemental Figure S4 is excellent and shows the appropriate insensitivity to time-step.

I have no further concerns and recommend the paper for publication.

R1) We thank the reviewer for their comments.

Reviewer #3 (Remarks to the Author):

C2) All my concerns have been addressed. I congratulate the authors to this really nice manuscript.

R2) We thank the reviewer for their comments.